# A mechanoelectrical mechanism for detection of sound envelopes in the hearing organ

Alfred L. Nuttall[1], Anthony J. Ricci[2,3], George Burwood[1], James M. Harte[4], Stefan Stenfelt[5], Per Cayé-Thomasen[6], Tianying Ren[1], Sripriya Ramamoorthy[7], Yuan Zhang[1], Teresa Wilson[1], Thomas Lunner[8,9], Brian C. J. Moore[10] & Anders Fridberger [1,5]

To understand speech, the slowly varying outline, or envelope, of the acoustic stimulus is used to distinguish words. A small amount of information about the envelope is sufficient for speech recognition, but the mechanism used by the auditory system to extract the envelope is not known. Several different theories have been proposed, including envelope detection by auditory nerve dendrites as well as various mechanisms involving the sensory hair cells. We used recordings from human and animal inner ears to show that the dominant mechanism for envelope detection is distortion introduced by mechanoelectrical transduction channels. This electrical distortion, which is not apparent in the sound-evoked vibrations of the basilar membrane, tracks the envelope, excites the auditory nerve, and transmits information about the shape of the envelope to the brain.

[1] Oregon Hearing Research Center, Oregon Health & Science University, Portland, OR 97239, USA. [2] Department of Otolaryngology-Head and Neck Surgery, Stanford University School of Medicine, 300 Pasteur Drive, Edwards Bldg., Stanford, CA 94025, USA. [3] Department of Molecular and Cellular Physiology, Stanford University School of Medicine, Stanford, CA 94025, USA. [4] Interacoustics Research Unit, DGS Diagnostics A/S, Technical University of Denmark, Ørsteds Plads Building 352, Room 117, DK-2800 Kgs.Lyngby, Denmark. [5] Department of Clinical and Experimental Medicine, Linköping University, SE 58183 Linköping, Sweden. [6] Department of Oto-rhino-laryngology, Head and Neck Surgery, and Audiology, F2074, Copenhagen University Hospital, Blegdamsvej 9, 2100 Copenhagen, Denmark. [7] Department of Mechanical Engineering, Indian Institute of Technology Bombay, Mumbai, Maharashtra 400076, India. [8] Eriksholm Research Centre, Oticon A/S, Rørtangvej 20, 3070 Snekkersten, Denmark. [9] Department of Behavioral Sciences and Learning, Linköping University, SE581 83 Linköping, Sweden. [10] Department of Experimental Psychology, University of Cambridge, Cambridge CB23EB, UK. These authors contributed equally: Alfred L. Nuttall, Anthony J. Ricci, George Burwood, James M. Harte, Stefan Stenfelt. Correspondence and requests for materials should be addressed to A.L.N. (email: nuttall@ohsu.edu) or to A.F. (email: anders.fridberger@liu.se)

Speech, music, and animal communication calls contain many different frequencies that change rapidly over time. Yet, spoken words can be recognized using only a limited amount of information about the slowly varying envelope of the stimulus[1–5]. A clear example of this comes from cochlear implant users, most of whom have excellent speech recognition when a few frequency bands of envelope information are presented through the implanted electrodes[6]. This information is conveyed to the auditory brainstem nuclei, where some cells respond selectively to specific rates of envelope modulation[7], and a systematic gradient of temporally specific neurons is found in one of the principal nuclei, the inferior colliculus[8]. This demonstrates that extraction of the envelope of sounds is essential for speech perception.

While it is clear that the pattern of action potentials in the auditory nerve reflects the shape of the envelope[9–12], frequency components corresponding to the envelope have not been found in the sound-evoked vibrations of the basilar membrane at the base of the cochlea[13,14]. But how can the auditory nerve convey information not present in the basilar membrane motion, which provides the stimulus that drives the nerve?

One proposed solution[15] starts from the observation that many natural sounds, including speech, contain multiple harmonics whose frequencies are integer multiples of some fundamental frequency. This may cause several harmonics to mechanically stimulate each inner hair cell, which would then respond preferentially at the peaks that result from the interactions among the harmonics. As a result, frequency components corresponding to the envelope would appear in the auditory nerve spike pattern. While this is an attractive idea, there are no experimental data that prove the theory.

Another potential mechanism for envelope detection relies on asymmetries in the currents generated by mechanically sensitive ion channels in auditory sensory cells. These channels have sigmoidal activation curves that cause receptor potentials to be dominated by inward currents. In mathematical models[16,17], such rectification may lead to envelope extraction if it is combined with low-pass filtering. Isolated hair cells can respond to stimuli with a fixed envelope[18,19] but it is not known whether changes in the envelope would be detected. Moreover, more recent modeling work emphasized neural mechanisms, such as rate adaptation in auditory nerve dendrites, as a mechanism for encoding envelopes[20].

To find the mechanism underlying envelope coding, we used an acoustic stimulus that allowed the envelope to be changed without altering the frequency content of the signal. This considerably facilitated interpretation of results. Using this stimulus, we recorded basilar membrane motion and hair cell receptor potentials, and performed experiments where cochlear potentials were recorded when auditory nerve activity was blocked. These experiments demonstrate that mechanically sensitive ion channels generate high-amplitude electrical potentials that correspond to the envelope of a complex stimulus. This process also produces electrical potentials at frequencies not present in the stimulus. Even though perception can sometimes result from such distortions[21], they are usually regarded as superfluous by-products of sensory transduction. In contrast, our data demonstrate that all distortions generated by the cochlea change when the envelope is altered.

## Results

**Acoustic stimulus**. To investigate the mechanisms underlying envelope extraction, we used stimuli with systematically changing envelopes but identical amplitude spectrum. To synthesize such sounds, three sine waves with equal amplitude and constant

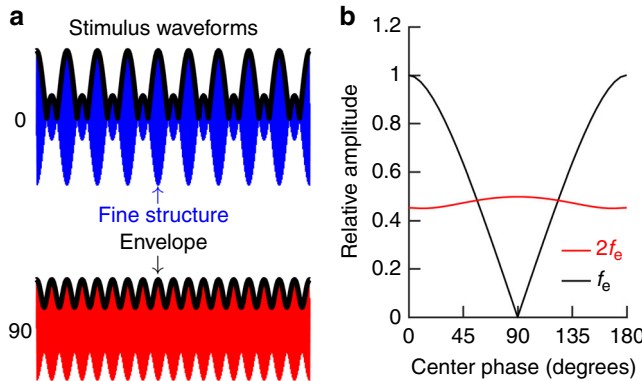

**Fig. 1** Acoustic stimuli. **a** Three tones with the same starting phase and identical frequency separation were added to produce the top blue waveform. In the lower red waveform, the center-tone phase was shifted by 90°, resulting in a flatter envelope. The envelope is marked with thick black lines. **b** Relative amplitude of envelope variations for different center-tone phases. The black line shows the amplitude at $f_e$ (also equal to the frequency difference between the three tones), which corresponds to the large peaks in the blue waveform of panel **a**. The amplitude at $2f_e$ corresponds to the peaks in the red waveform. The envelope amplitudes were computed by Fourier transformation of the magnitude of the Hilbert transform for each waveform

frequency separation were added:

$$X(t) = A\sin(2\pi f_1 t) + A\sin(2\pi f_2 t + \varphi) + A\sin(2\pi f_3 t) \quad (1)$$

$$f_2 = f_1 + f_e \quad (2)$$

$$f_3 = f_1 + 2f_e \quad (3)$$

Here, $f_e$ denotes the frequency difference between $f_1$ and $f_2$, $\varphi$ is the phase of the center tone, $A$ is the stimulus amplitude, and $t$ is time. When the three tones had the same starting phase, the envelope had a pattern of alternating small and large peaks (shown schematically by the top blue waveform in Fig. 1a). The large peaks recurred at a frequency equal to $f_e$. When the phase of the center tone was shifted by 90°, the large peaks were replaced by smaller ones, which had the frequency $2f_e$ (Fig. 1a, lower red waveform). Envelope shapes in between these extremes were generated by varying the center-tone phase over a 180° range.

The relative magnitude of the envelope fluctuations at $f_e$ and $2f_e$ is plotted in Fig. 1b as a function of center-tone phase. Note that the magnitude at $f_e$ declines to zero for a center-tone phase of 90°, whereas the magnitude at $2f_e$ remains nearly constant. These effects result solely from the superposition of waves with different relative phase, which means that all these stimuli have identical magnitude spectra. This distinguishes these sounds from 'ordinary' amplitude modulation, where alterations in the envelope are associated with changes in the level of the primaries.

Previous studies established a nonlinear relationship between the acoustic stimulus and the response of the hearing organ. When a stimulus with three components encounters such a nonlinearity, a Taylor series expansion may be used to model the effects[21,22]. Using the definitions in Eq. 1, the quadratic component of the series includes the term:

$$f^2(x) = 2 \cdot \cos(\varphi) \cdot \cos(2\pi f_e t) + \cos(2\pi 2f_e t)\ldots . \quad (4)$$

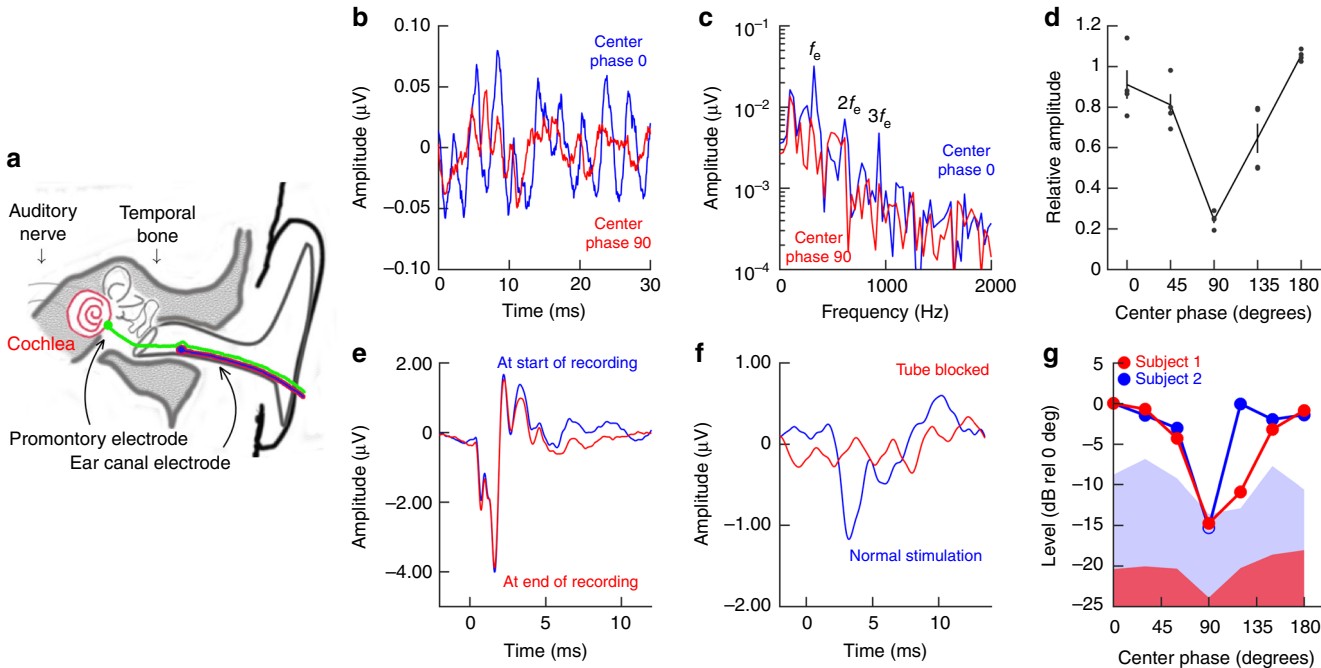

**Fig. 2** Human responses to three-tone acoustic stimuli. **a** Schematic diagram of the human temporal bone, showing the positions of the recording electrodes. **b** Examples of ear canal recordings for center-tone phase 0° (blue) and 90° (red). Each waveform is formed by averaging responses to 25,000 condensation three-tone bursts, with 25,000 rarefaction bursts, at a stimulus level of 84 dB SPL. A 3rd-order high-pass filter with 100 Hz cutoff frequency was applied to reduce low-frequency noise. **c** Magnitude spectra of the responses shown in **b**. **d** Normalized average responses at $f_e$, 300 Hz, for four subjects. Vertical lines denote the sem, and dots represent individual data points. **e** Compound action potential and summating potentials recorded from the cochlear promontory at 90 dB normal Hearing Level (nHL), at the start and end of the recording session. **f** Blocking the acoustic stimulus tube abolishes compound action potential responses from the promontory. Stimuli were 4 kHz tone bursts at 90 dB nHL. **g** Normalized responses as a function of the center-tone phase for two subjects. Noise levels for each subject are given by the red and blue colored areas. A permutation test[52] was used to determine that each data point was statistically separated from the noise. For phases 0, 30, 60, 120, 150, and 180°, all $P$ values were <0.00345, meaning that the probability that these responses were false positives was less than 4 in 1000. For the 90° phase, the data point for subject 2 was not significantly different from the noise. $f_e$, frequency of envelope variations; $2f_e$, component at twice $f_e$

The model thus predicts that envelope-following responses would occur when listening to the three-tone stimulus described above, and may give useful information about the properties of these responses.

**Envelope responses of the human ear**. To determine whether these three-tone stimuli are relevant for investigating envelope coding in the human ear, we recorded electrophysiological responses from four subjects, using electrodes positioned close to the tympanic membrane (schematic in Fig. 2a). A relatively flat stimulus envelope, with center-tone phase of 90°, resulted in a smooth response waveform (the red trace in Fig. 2b shows an example recording from one subject). A switch to a peakier envelope (center-tone phase 0°) produced additional components, superimposed on the smooth response (blue trace in Fig. 2b). These additional components were evident in the amplitude spectrum as peaks at $f_e$ and $2f_e$ (blue trace in Fig. 2c). The amplitude of the $f_e$ peak depended on the phase of the center tone (Fig. 2d; means ± standard error of the mean, sem; $p = 2.6 \times 10^{-6}$, linear mixed model; $n = 4$), but this was not the case for the $2f_e$ peak ($p = 0.76$, linear mixed model; the normalized mean amplitude ± sem at center-tone phases 0°, 45°, 90°, 135°, and 180° was $1.04 \pm 0.12$; $1.16 \pm 0.08$; $0.94 \pm 0.11$; $0.82 \pm 0.19$; $1.04 \pm 0.09$, respectively).

Electrical potentials recorded in the ear canal are influenced by potentials generated in the cochlea, auditory nerve, and various brainstem nuclei. To better isolate a cochlear component, electrodes were placed on the promontory (Fig. 2a, green electrode), an invasive procedure that is possible in only a few

cases, where cochlear function is continuously monitored during surgery. In two consenting patients undergoing surgery for superior semi-circular canal dehiscence, promontory electrodes were used to record responses to brief click-like sounds, which cause synchronous activation of many auditory nerve fibers (Fig. 2e). The similarity between responses recorded at the beginning and at the end of the recording session (cf. blue vs. red trace in Fig. 2e) is evidence that the electrode maintained its position on the promontory throughout the recording. For both patients, 4 kHz tone bursts resulted in reproducible responses that were abolished when the loudspeaker tube was blocked (Fig. 2f). After these controls, responses to the three-tone stimulus were recorded while systematically varying the phase of the center tone. The response amplitude at $f_e$ was dependent on the center-tone phase (Fig. 2g; a permutation test verified that each data point was significantly different from the system noise level, which is depicted by the blue and red fields in the graph. The only exception was the 90° response for subject 2). In subject 1, the amplitude at the $2f_e$ frequency was flat across center-tone phases. In subject 2, where the noise level was higher, the amplitude fell by 8 decibels (dB) for center-tone phase 90°. Taken together, this demonstrates that the three-tone stimulus is appropriate for investigating envelope coding in humans.

**Organ of Corti electrical signals track the envelope**. To find the mechanism underlying the envelope coding shown in Fig. 2, we stimulated the ears of deeply anesthetized guinea pigs with the three-tone stimuli while measuring basilar membrane motion with laser Doppler vibrometry (Fig. 3a and ref.[23]). If the basilar

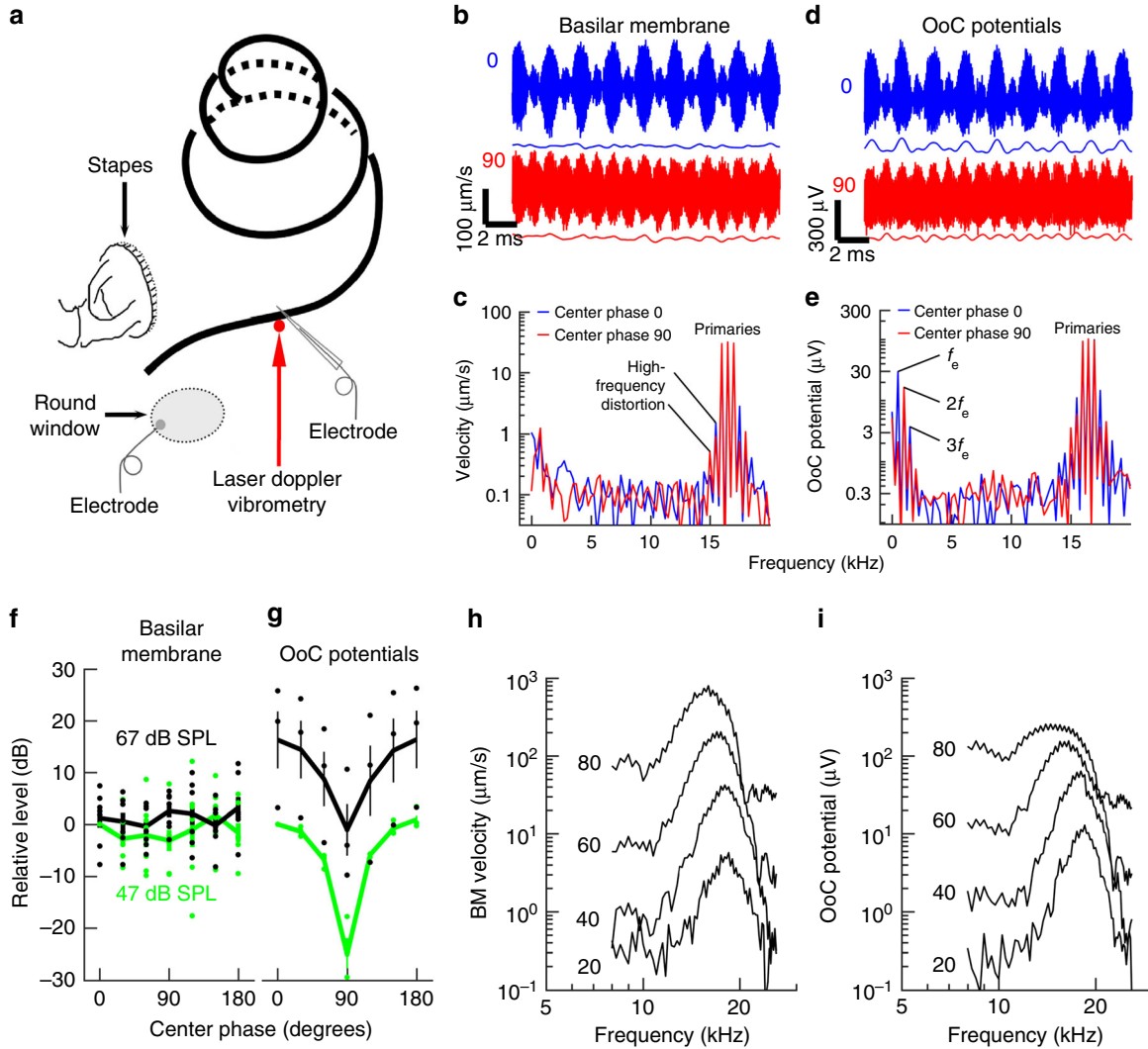

**Fig. 3** Envelopes and their effect on distortion. **a** Acoustic inputs reached the sensory cells through the stapes and the resulting vibrations of the basilar membrane (black spiraling line) were measured with laser Doppler vibrometry. Optical coherence tomography was used to measure organ of Corti displacement through the intact round window membrane. Electrical potentials produced by the hair cells were recorded with an electrode positioned inside the hearing organ or at the round window. **b** Examples of responses recorded from the basilar membrane in response to the three-tone complexes for center-tone phase 0° (blue) and 90° (red). The stimulus frequencies were 16, 16.5, and 17 kHz; the largest response of the recording location was at 17.5 kHz and the stimulus level was 67 dB sound pressure level, SPL, relative to 20 µPa. The thin lines are low-pass filtered versions of each waveform (filter cutoff frequency, 3 kHz). **c** Spectra of the data in panel **b** reveal strong responses to the three primary tones, as well as high-frequency distortion components flanking the primaries. Amplitudes at $f_e$ nd $2f_e$. were at the system noise level. **d**, **e** Electrical responses to the stimuli shown in Fig. 1a, measured by a calibrated electrode placed inside the organ of Corti (OoC). Thin lines are low-pass filtered responses, 3 kHz cutoff frequency. **f** The magnitude of basilar membrane motion at 500 Hz ($f_e$ was unaffected by the center-tone phase (mean ± sem; $n = 7$), dots denote individual data points. Green color is used for data at 47 dB SPL, black color for 67 dB SPL. **g** Levels of organ of Corti electrical potentials at 500 Hz depended strongly on center-tone phase (mean ± sem, $n = 4$ at 47 dB SPL; $n = 3$ at 67 dB SPL). Color code identical to panel **f**. **h**, **i** Tuning curves of the basilar membrane's motion and organ of Corti electrical potentials. Numbers next to each curve denote stimulus levels in dB SPL. $f_e$, frequency of envelope variations; $2f_e$, component at twice $f_e$

membrane responded to the envelope, the stimulus shown in the blue waveform in Fig. 1a (center-tone phase of 0°) would produce a response with significant amplitude at the frequency corresponding to the repetition rate of the large peaks ($f_e$). The response recorded from the basilar membrane (Fig. 3b, top blue graph, 60 averages) resembled the stimulus waveform, but Fourier transformation revealed that the amplitude at $f_e$ was only 0.27 µm s$^{-1}$ (Fig. 3c, blue curve; the amplitude at $2f_e$ was also 0.27 µm s$^{-1}$). These values are within the noise floor of the measurement, a result that also can be appreciated from the fact that a low-pass filtered version of the trace (thin blue line in Fig. 3b) showed fluctuations with no consistent pattern.

The stimulus shown by the red trace in Fig. 1a has a relatively flat envelope. In response to such a stimulus, the basilar membrane vibration amplitude at $f_e$ was 0.56 µm s$^{-1}$ and the amplitude at $2f_e$ was 0.11 µm s$^{-1}$ (Fig. 3c, red curve and red thin line), values that are within the noise floor of the measurement. These results, which are consistent with previous studies[13,14], show that no component corresponding to the envelope could be detected in basilar membrane vibrations.

A strikingly different picture emerged when a microelectrode with ~1 µm tip diameter was advanced into the hearing organ and placed close to the sensory outer hair cells. This electrode, which recorded the response of a small group of cells around its tip[24],

revealed asymmetric electrical potentials with larger excursions in the positive direction (Fig. 3d). The low-pass filtered version of this trace (thin blue line in Fig. 3d) revealed peaks that followed the envelope of the stimulus. This was also reflected in the spectrum of the response (Fig. 3e), which showed that high-frequency distortion was present, but also a prominent peak at $f_e$ (Fig. 3e), along with smaller peaks at $2f_e$ and $3f_e$. With a peaked envelope (Fig. 3e; blue lines, center phase of 0°), the average level at $f_e$ was 7 ± 4.3 dB below the level of the first primary tone ($n = 4$; mean ± sem), but the level fell when the envelope was flatter (center phase of 90°; red trace; all data were corrected for the low-pass filtering inherent to the glass electrode, see ref.[24]). For center-tone phase 0°, the peak at $2f_e$ was 9.7 ± 4.6 dB below the level of the first primary tone; the corresponding value for center-tone phase 90° was 10.3 ± 5.4 dB. These data show that the hearing organ generates electrical signals that track the envelope. Since such signals were not present in the output of the loudspeaker or detected in basilar membrane vibrations, these results suggest that they are a result of processing within the organ of Corti.

To further characterize this envelope-tracking signal, we systematically altered the phase of the center component of the stimulus. At the basilar membrane, no signal at either $f_e$ or $2f_e$ emerged from the noise despite 60 or 120 averages (Fig. 3f shows averaged results at $f_e$, 500 Hz, $n = 7$), but changes in center-tone phase affected the organ of Corti potentials, which gradually changed as the envelope moved from the peaky shape to the flatter one. The resulting curve (Fig. 3g) resembled the plot of the relative amplitude of the envelope variations (Fig. 1b). A flat envelope resulted in 23 ± 3 dB smaller levels at $f_e$, relative to levels recorded with a peaked envelope (Fig. 3g; $n = 4$; 47 dB sound pressure level, SPL). This effect was somewhat reduced at higher stimulus level (18.5 ± 2 dB difference between the 0 and 90° phases at 67 dB SPL; $n = 3$). The spectral peak at $2f_e$ did not depend on the phase of the center tone (normalized amplitude for phase 0°, 2.39 ± 1.21 dB; amplitude at phase 90°, 2.85 ± 0.4 dB). The changes in the organ of Corti potentials were statistically significant for the $f_e$ peak ($p = 2.1 \times 10^{-11}$, linear mixed model), but this was not the case for alterations in basilar membrane vibrations ($p = 0.26$, linear mixed model).

To verify that the data in Fig. 3 came from normally functioning hearing organs, frequency-tuning curves were recorded. The basilar membrane responded to low-level sounds and showed sharp tuning and compressive nonlinearity (Fig. 3h), all of which characterize normal cochleae. Nonlinearity was more pronounced in electrical potentials, where a 60-dB stimulus level increase resulted in only a 22-dB response change (Fig. 3i; basilar membrane data in Fig. 3h were acquired after electrode penetration, which induced a 9-dB loss of auditory sensitivity).

**Envelope responses are undetectable at the basilar membrane**. In the cochlea, electrical and mechanical events are tightly linked. Hence, it is surprising that the electrical envelope-following responses shown in Fig. 3 were not apparent in the vibrations of the basilar membrane. To further explore this phenomenon, we measured sound-evoked displacements using optical coherence tomography (OCT). This interferometric technique produced images of the hearing organ (Fig. 4a) where the basilar membrane and the top of the sensory cells, the reticular lamina, could be identified and their response to sound stimulation measured. The noise floor at 500 Hz was 0.06–0.3 nm (Fig. 4b–e), implying that mechanical events occurring near the threshold of audibility would be detectable[25].

The data in panels 4b, c are from an animal with a 2-dB loss of auditory sensitivity at the time of recording (as reflected in

measurements of compound action potentials). The reticular lamina showed a 0.24-nm peak at $f_e$ (Fig. 4b), but this component did not emerge from the noise at the basilar membrane (Fig. 4c). High-frequency distortion products were however present on both structures (the insets in Fig. 4b, c shows the frequency region around the primaries at expanded scale, where the high-frequency distortion is evident as the peak on the right of the three primaries). Envelope-tracking responses were found at the reticular lamina in 5 out of 8 sensitive preparations at 74 dB SPL, but in no case could such components be detected in the basilar membrane's motion.

Since the envelope-following mechanical responses were close to the noise floor at 74 dB SPL, the stimulus level was increased by 20 dB. This resulted in a 0.7-nm peak at the reticular lamina (Fig. 4d) but again, no envelope-tracking response was detected at the basilar membrane (Fig. 4e; note the low basilar membrane noise floor in this preparation, which had fully intact compound action potential thresholds at the time of recording. As shown in the insets, both structures showed high-frequency distortion products).

Using a metal electrode positioned on the round window membrane (Fig. 3a), electrical responses to the three-tone stimulus were recorded following the OCT recordings. As seen in Fig. 4f, the envelope signal dominated the response at 74 dB SPL. In addition to the envelope signal, several low-frequency peaks that lacked detectable counterparts at either the basilar membrane or the reticular lamina were evident. Furthermore, the data in Fig. 5 show that electrical envelope-following responses were present at the round window membrane at 44 dB SPL.

To summarize, mechanical responses at $f_e$ were present at moderate and high stimulus levels at the reticular lamina, but these signals could not be detected at the basilar membrane despite extensive averaging and noise floors sometimes better than 0.1 nm. Responses at $2f_e$ were detected at neither the basilar membrane nor the reticular lamina, but this frequency component was prominent in the round window recordings.

**Envelope-tracking signals are generated by sensory cells**. To further probe the properties of the electrical envelope-following response, the metal electrode on the round window was used to record 'far-field' electrical responses from sensory cells and neurons.

A pattern similar to the one in the organ of Corti potentials was evident. With a peaked envelope (center-tone phase 0°), the level at $f_e$ was 19 ± 2 dB higher than the levels recorded with a flatter stimulus envelope (center-tone phase 90°; Fig. 5a; 44 dB SPL). The average magnitude at $2f_e$ was 1.8 ± 0.5 dB higher for phase 90° than it was for phase 0°, consistent with the theoretical curve shown in Fig. 1b. When the stimulus level was increased by 20 dB, response magnitudes increased but the tip-to-tail ratio was similar (21 ± 3 dB). The effect of center-tone phase was significant ($p = 1.4 \times 10^{-56}$; $n = 13$, linear mixed model).

In the experiments shown in Fig. 3, the electrode was placed inside the organ of Corti. The recorded signals are dominated by a small number of outer hair cells (electrode space constant <50 μm, ref.[24]), but the round-window electrode records the response of a larger group of cells, including afferent neurons. To assess contributions from the auditory nerve, we silenced its action potentials by applying the sodium-channel blocker tetrodotoxin (TTX) directly to the round window membrane (1 μl of a 0.5 mM TTX solution, producing a 40-dB decrease in the amplitude of the compound action potential evoked by tone bursts).

Consistent with previous reports[26], TTX caused a small change in the amplitude of electrical responses across frequencies. To

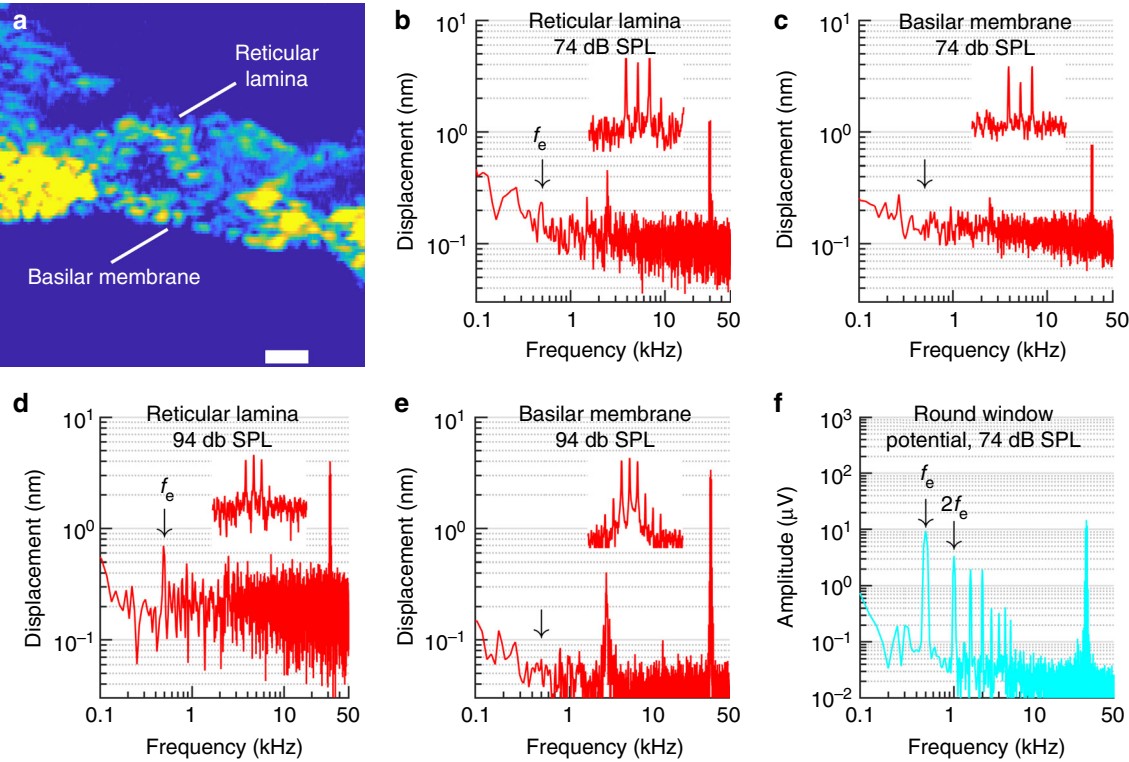

**Fig. 4** No envelope tracking at the basilar membrane. **a** Structural optical coherence tomography image of the organ of Corti. Scale bar, 50 μm. **b** Spectrum of reticular lamina displacement in response to a three-tone stimulus at 74 dB SPL with center-tone phase 0°. Note the envelope signal, $f_e$ barely rising above the noise floor. The peak at ~2.5 kHz was caused by noise within the recording system, and the response near the three primaries at 29.5, 30, and 30.5 kHz is shown in greater detail in the inset. **c** Spectrum of basilar membrane displacement for the same acquisition as panel **b**. The arrow marks the frequency of the expected envelope signal, which was not detected. Inset is plotted with the same parameters as in panel **b**. **d** Reticular lamina displacement spectrum at 94 dB SPL. Primaries at 31, 31.5, and 32 kHz; center-tone phase 0°. Inset shows the region around the three primaries. **e** Basilar membrane displacement spectrum for the same acquisition as panel **d**. Inset has the same parameters as in **d**. **f** Prominent envelope signals were detected in electrical potentials recorded at the round window membrane at the completion of vibration measurements. Stimulus level, 74 dB SPL. $f_e$, frequency of envelope variations; $2f_e$, component at twice $f_e$

normalize for this change, we calculated the tip-to-tail ratio, as defined in Fig. 5a, and found that it was unaffected ($p = 0.86$; linear mixed model; Fig. 5b; $18 \pm 1.5$ dB tip to tail ratio at 44 dB SPL and $19 \pm 2.3$ dB ratio at 64 dB SPL). This indicates that auditory nerve activity does not cause the envelope-tracking electrical potential, but rather reflects it.

Recordings of round window electrical potentials were also used to examine the relation between envelope-tracking responses and the frequency spacing between the tones in the stimulus. The largest tip-to-tail ratios were observed for frequency separations smaller than 500 Hz (Fig. 5c; tip-to-tail ratio at 100 Hz, $21 \pm 1.4$ dB at 44 dB SPL, and $24 \pm 1.3$ dB at 64 dB SPL); TTX had no influence on the ratios (Fig. 5d; $p = 0.19$, linear mixed model).

Since blocking auditory nerve activity left the envelope-tracking electrical potential intact, we conclude that it was generated by the sensory hair cells.

**Envelope effects on distortions.** Apart from the low-frequency distortions described above, an acoustic stimulus with frequency components at 17, 17.5, and 18 kHz produces high-frequency intermodulation distortion, for instance at 18.5 kHz (2f3-f2, where f2 and f3 are the frequencies of the center and highest tones). The data shown in Fig. 3c and e suggest that the envelope affects the magnitude of these high-frequency distortions. Indeed, when basilar membrane vibration amplitudes were plotted as a function of center-tone phase, the magnitude at 2f3-f2 was found to be $13 \pm 2.2$ dB lower when the envelope was flat (center-tone

phase of 90°), than when it was peaked (center-tone phase of 0° or 180°; Fig. 6a, $n = 7$, 64 dB SPL). This effect was slightly more pronounced in organ of Corti electrical potentials ($17 \pm 4.4$ dB tip-to-tail ratio, $n = 3$). The dependence on center-tone phase was statistically significant for both the basilar membrane and organ of Corti potentials ($p < 0.01$ in both cases, linear mixed model).

The stimulus envelope also affected other high-frequency distortion components. At 3f1-2f2 (Fig. 6b), minima were observed for phases 0° and 180°, with a broad peak near 90° (tip-to-tail ratio $10 \pm 1.2$ dB on the basilar membrane and $12 \pm 4.5$ dB in the organ of Corti; $p < 0.001$ for both effects; linear mixed model). The tip-to-tail ratio at 2f1-f2 was smaller (Fig. 6c), but the phase effect was nonetheless statistically significant ($p = 0.02$; linear mixed model). Limited data acquired at 44 dB SPL showed the same pattern (Fig. 6d). Hence, the shape of the envelope affected all high-frequency distortion products that we were able to record.

**Transduction channels generate envelope-tracking responses.** The data shown above demonstrate that the hair cells generated a local electrical signal that tracked the envelope of the acoustic stimulus. To determine the mechanism behind this effect, we used the patch-clamp method to record currents evoked by deflections of inner hair cell stereocilia. In response to step deflections, an initial inward current was followed by gradual adaptation (Fig. 7a, top graph). Plotting the normalized maximal current as a function of bundle displacement revealed the sigmoid

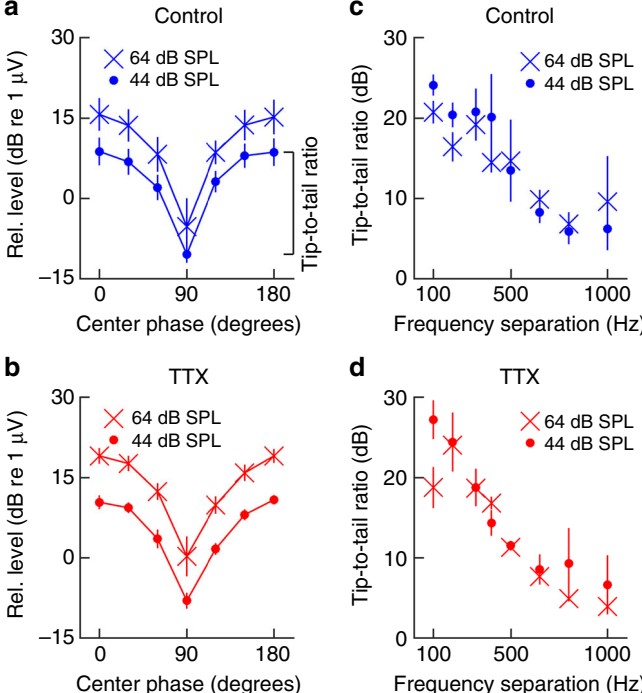

**Fig. 5** Envelope tracking in cochlear potentials. **a** Round window electrical signals track the amplitude of envelope variations. The three tones were separated by 320 Hz for this recording. Data points represent the mean ± sem from 13 animals at 44 dB SPL and 4 animals at 64 dB SPL. **b** Envelope tracking remained after application of tetrodotoxin (TTX). **c** Tip-to-tail ratios in control animals. **d** Tip-to-tail ratios were similar after TTX

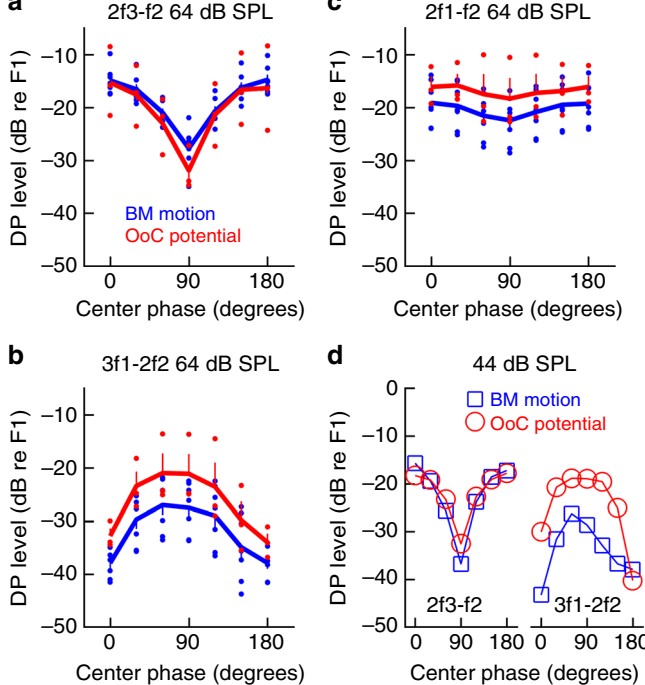

**Fig. 6** High-frequency distortion depends on center-tone phase. **a** The amplitude of the 2f3-f2 distortion product depends on the phase of the center tone. Similar findings were evident in both basilar membrane (BM; $n = 7$; blue) vibrations and organ of Corti (OoC; $n = 3$; red) potentials. **b** Corresponding data for the 3f1-2f2 distortion product. **c** Smaller effects of center tone phase were evident at 2f1-f2. **d** Data from a single animal with successful recording of high-frequency distortion at 44 dB SPL. In panel **a**–**c**, vertical bars denote the sem, and the lines are drawn through the mean values. Dots denote individual data points

relationship expected from normally functioning hair cells (Fig. 7a, bottom graph). After verifying the presence of normal hair cell responses, the three-tone complex with a systematically varying center-tone phase was used. This stimulus produced asymmetric responses dominated by inward currents, and an obvious response change when the center-tone phase moved from 0 to 90° (Fig. 7b). To quantify these changes, the amplitude spectrum of the response was computed (Fig. 7c). At center-tone phase 0°, a 17-pA peak appeared at $f_e$ (100 Hz). Its amplitude declined to 0.51 pA at center-tone phase 90°, a 30-dB change that brought the envelope signal to the noise floor of the recording system (the envelope also affected high frequency distortion, as shown in Supplementary Fig. 1).

To examine whether this response was frequency dependent, we varied the center frequency of the three-tone complex over a 1400-Hz range, from 400 to 1900 Hz, while keeping the spacing between the three tones constant at 100 Hz. The hair cells continued to produce responses at $f_e$ throughout this frequency range, with the V-shaped dependence on center-tone phase described above (Fig. 7d; at $2f_e$ the average amplitude was 2.4 ± 0.4 dB higher for center-tone phase 90° than it was at center-tone phase 0°). A fluid jet stimulating device was also used to account for any effect of hair bundle loading and similar responses were observed as regards the phase shift and envelope tracking. Stimulus magnitudes were compared using stiff probe or fluid jet and comparable results were obtained for stimuli evoking 20–80% of the maximal current response. The absolute magnitude of the stimulation varied based on stimulus modality and stiff probe shape as predicted. Given that these data were collected using voltage-clamp where the cells were clamped to −84 mV, no effects on voltage-gated channels were expected or observed.

To further probe the underlying mechanisms, we constructed a mathematical model based on the properties of mechanically

sensitive ion channels. The relation between displacement, $X$, and the receptor current, $I$, is described by (review, ref.[27]):

$$I(X) = \frac{I_{\max}}{1 + e^{\left(-\frac{Z(X-X_0)}{k_b T}\right)}} \quad (5)$$

where $Z$ is the single-channel gating force, $k_b$ is Boltzmann's constant and $T$ is the absolute temperature. $X_0$ shifts the function horizontally, determining the current that flows into the cell at rest[28,29] (Fig. 7e). For the three-tone stimulus, $X$ is given by Eqs. (1)–(3). In the model, the stimulus was applied to the stereocilia with 1-nm displacement amplitude at each frequency, which corresponds to a moderately intense acoustic stimulus[30].

With a peaked envelope (center-tone phase 0°), the model's receptor current contained a component at $f_e$ (500 Hz, blue waveform and peak in Fig. 7f). This component, which was absent from the stimulus, decreased in level by 60 dB when the envelope became flatter (center-tone phase 90°, red trace in Fig. 7f, see also Supplementary Fig. 2). Currents at $f_e$ were always generated (Fig. 7g), except when $X_0$ was exactly equal to zero, which brought the resting open probability of the transduction channels to the value of 0.5. Although the in vivo resting open probability is unknown, isolated hair cells show values in the range 0.28–0.46 (ref.[27]) which lends credence to this aspect of the model.

When $X_0$ deviated from zero, the model also generated currents at $2f_e$ (orange peak in Fig. 7f) but the amplitude of this frequency component showed little dependence on center-tone phase (less than 0.2 dB change when the center-tone phase moved from 0° to 90°).

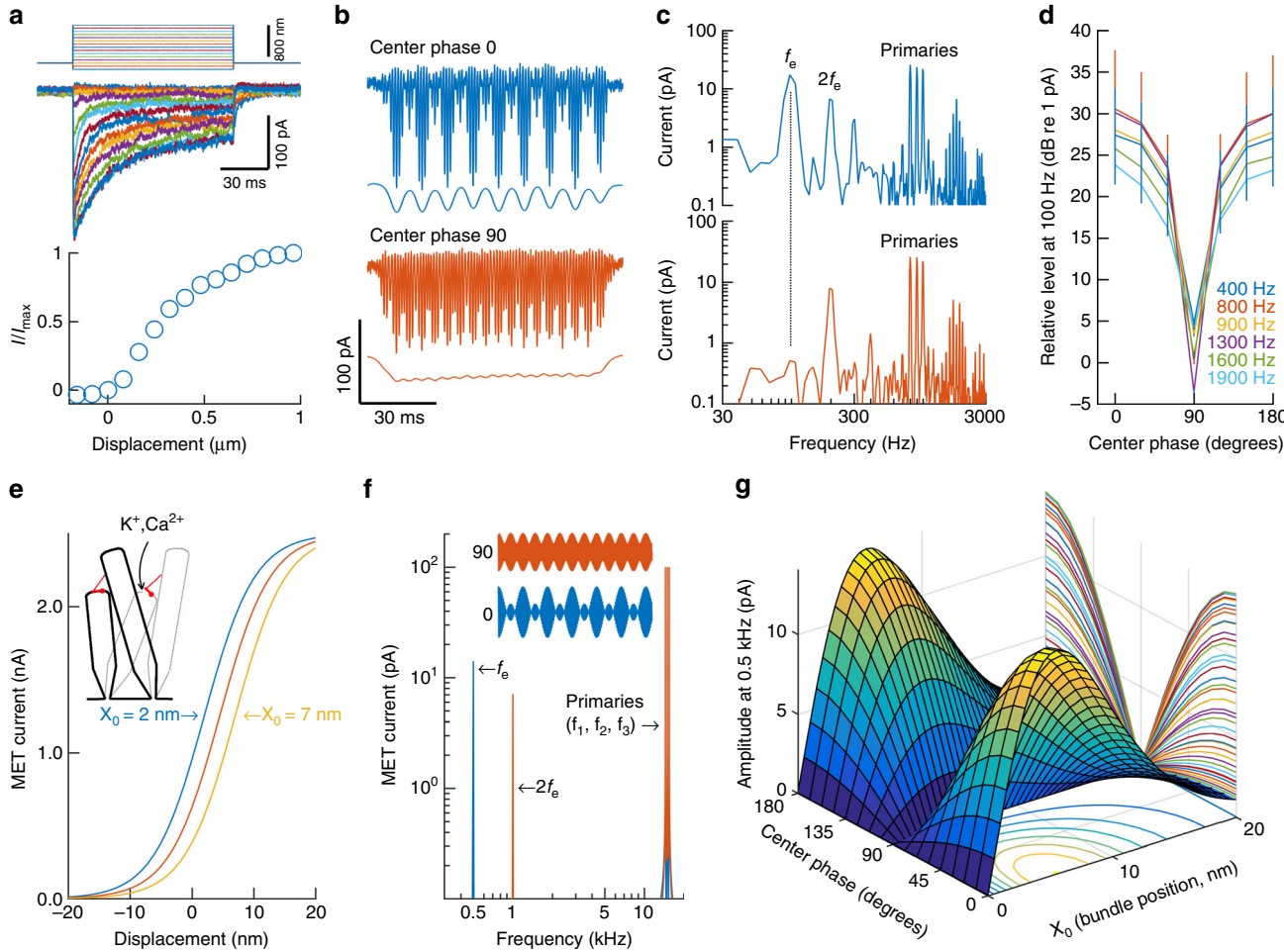

**Fig. 7** Origin of the envelope signals. **a** When pushed sideways, electrical currents flow into sensory cell stereocilia as mechanically sensitive ion channels open. These currents have a sigmoidal relation to the bundle displacement (lower graph). **b, c** Example hair cell currents evoked by three-tone stimulation of stereocilia, using a stiff stimulus probe. The thin lines are low-pass filtered versions of each trace (filter cutoff, 500 Hz). At center-tone phase 0°, the magnitude spectra (**c**) revealed peaks at $f_e$ and $2f_e$. The $f_e$ peak disappeared at center-tone phase 90°. **d** Averaged data from 10 cells in 10 different animals (±sem). The frequencies represent the center frequency of each stimulus. **e** Hair cell mechanoelectrical transduction channels have sigmoidal activation curves described by first-order Boltzmann functions. The sideways shift of the curves is a consequence of adaptation and is described by the model parameter $X_O$. **f** Frequency spectra of model receptor currents evoked by three-component stimuli. A large peak at $f_e$ is observed when the center-tone phase is zero (blue waveform); this peak is abolished at center-tone phase 90° (red waveform). The peak at $2f_e$ corresponds to the 1-ms periodicity present regardless of center phase. Parameters: $I_{max}$, 2.5 nA; $X_O$, 5.5 nm; gating force, $Z$, 1.05 pN; Temperature, 310.15 K; $K_b$, 1.381 × 10$^{-23}$ J K$^{-1}$; stimulus frequencies ($f_1, f_2, f_3$), 14.5, 15, 15.5 kHz; stimulus amplitude, 1 nm. The model contains no temporal parameters, hence assuming MET channels are infinitely fast. **g** At $X_O = 0$ no envelope coding is possible. As soon as the resting position of the stereocilia deviates from this value, the receptor current contains a signal corresponding to the envelope. The maximum is at 7 nm. At large values of $X_O$, the overall amplitude of the receptor current is reduced, because this causes the stimulus to be applied near the flat portion of the Boltzmann function, where the slope is small. Except for $X_O$, parameters identical to those for panel **f** were used. $f_e$, frequency of envelope variations. $2f_e$, component at twice $f_e$

Compared to the primaries, the amplitude of the model's $f_e$ peak was smaller (−17 dB for $X_0 = 5.5$ nm, corresponding to 0.2 open probability) than experimentally observed values (−7 ± 4.3 dB; Fig. 3e; see also Fig. 7c). Hence, additional nonlinearities are necessary to fully match the experimental results. These nonlinearities may reside in bundle mechanics[18,19,31,32], or result from feedback within the organ of Corti[33,34]. We conclude that the sigmoidal activation curve of mechanically sensitive ion channels generated currents that extract the envelope of complex harmonic stimuli.

## Discussion

Here we examined the mechanism used by the inner ear to encode critical features of communication-relevant sounds. When such complex stimuli arrive in the cochlea, they cause deflection of

stereocilia on auditory sensory cells, whose mechanically sensitive ion channels generate electrical currents that track the stimulus envelope. In our patch-clamp experiments, the amplitude of the envelope-tracking currents was close to that for the primaries. The amplitude of the envelope-tracking electrical response was also large in microelectrode recordings from within the organ of Corti, and recordings of electrical potentials at the round window showed that pharmacological block of auditory nerve activity had no effect on envelope coding. Hence, we conclude that the main mechanism for envelope detection is the generation of distorted electrical potentials by the sensory hair cells. These potentials excite the auditory nerve, which informs the brain about the shape of the envelope.

At the basilar membrane, neither OCT nor laser vibrometry detected signals corresponding to the envelope. Since such signals were present at the reticular lamina at moderate and high

stimulus levels, a mechanical filtering process within the organ of Corti is evident. Support for such complex micromechanics has emerged from several recent studies[30,35]. While it may be argued that an envelope signal would be detected at the basilar membrane if the noise floor was even better, we note that the noise floor in Fig. 4e was about 0.06 nm, and that previous recordings showed that tones at 10 dB SPL evoked 0.09-nm responses[25].

Although the hearing organ has long been known to generate distortion products[21], which are useful for diagnostic purposes[36,37], they are generally viewed as by-products of sensory transduction and nonlinear basilar membrane motion[38,39]. Previous measurements of high-frequency distortions established that their amplitudes increased as the separation between the stimulus frequencies became smaller[40]. This is noteworthy, because many behaviorally relevant sounds are harmonic complexes with small separation between components[41]. Here we demonstrated that the amplitudes of all of the distortion components that we were able to record, whether high or low in frequency, depended on the shape of the envelope (Fig. 6a–d), an effect that has not previously been described but may be perceptually important.

The present recordings were performed with tone complexes near the best frequency of the recording location. Because of the small space constant of electrodes placed inside the organ of Corti ($<50 \mu m$[24]), and the restricted region of the basilar membrane excited by these tone complexes, the electrical distortions we recorded are a local phenomenon, where each tone complex resulted in the excitation of only a small number of nerve fibers. Although the underlying mechanism was previously unclear, envelope responses are therefore useful for estimating the tuning of the hearing organ[11,12]. Since frequency components corresponding to the envelope could be detected at the reticular lamina only at moderate and high stimulus levels (Fig. 4), but never in the vibrations of the basilar membrane at any of the stimulus levels we employed (Figs. 3, 4, refs;[13,14]. see also ref.[42]), it appears that mechanical responses at the frequency of the envelope variations do not contribute significantly to the encoding of the envelope of low-level stimuli. Instead, the envelope is encoded mainly through electrical distortion generated by the hair cells, which allows information about the envelope of high-frequency signals to be transmitted to the brainstem despite the limited bandwidth of the auditory nerve. This is somewhat reminiscent of the demodulation to baseband signal processing used in telecommunications systems.

When listening to closely spaced tones at the frequencies f1 and f2 (f2 > f1), most people are able to hear additional tones that are not physically present. Some of these combination tones are perceived as tones with specific frequencies, the most easily heard one having a frequency of 2f1-f2. The perceived magnitudes of the high-frequency combination tones produced by three primaries are influenced by the relative phases of the stimulus tones[43,44], consistent with the data presented above. However, listeners do not usually hear a tonal component corresponding to the envelope repetition rate (f2-f1), except at high sound levels. It is possible that perception of the f2-f1 component is correlated with the appearance of mechanical envelope components at the reticular lamina, while the strong hair-cell generated electrical signal at this frequency may contribute to the internal representation of the envelope and to the perception of the pitch of complex sounds, including the missing fundamental (e.g., ref.[45]).

## Methods

**Human experiments**. Human experiments were approved by the ethics review boards in Linköping, Sweden, and Copenhagen, Denmark. Four normal-hearing consenting volunteers (3 males and 1 female, ages 31–48), comfortably reclined in an electrically shielded sound-proof room, participated in the first set of experiments. Under visual control, a gold-foiled insert earphone electrode was positioned inside the external ear canal, as close as possible to the tympanic membrane. The ground electrode was attached to the mastoid process on the other side and a reference electrode on the forehead. These electrodes were connected to an Eclipse EP25 recording system (Interacoustics A/S, Middelfart, Denmark) that also generated the acoustic stimuli, which were delivered to the subjects through insert earphones (EarTone 3A, 3M Inc, St Paul, MN, USA). The stimuli were 30-ms tone complexes with components at 3750, 4062.5, and 4375 Hz and a 1-ms cos-squared rise/fall time, repeated 32.7 times per second. Fifty thousand individual responses were averaged, and the frequency components at $f_e$ and $2f_e$ were extracted using the fast Fourier transform.

In a second set of human experiments, recordings were made from the cochlear promontory in two consenting subjects undergoing surgery for superior canal dehiscence. A bayonet forceps was used to advance a sterilized sub-dermal needle electrode through the posterior-inferior quadrant of the tympanic membrane until contact was made with the cochlear bone. To hold the electrode securely in place while delivering acoustic stimuli, a compressed insert earphone was inserted deeply into the ear canal, an approach that also facilitated stimulus calibration. To create a differential recording configuration, a second electrode was placed on the contralateral mastoid process, while a sub-dermal needle electrode on the contralateral cheek served as the ground. Click and tone burst levels were 90 dB normal hearing level (nHL), and the 4 kHz tone bursts had a Blackman window. The three-tone stimuli were delivered at 80 dB SPL and had 100-ms duration. An Interacoustics EP25 system was used for stimulus generation and response recording, with a sampling rate of 30 kHz.

**In vivo recordings in guinea pigs**. Young guinea pigs weighing <350 g were prepared for physiological recordings using procedures approved by the Oregon Health and Science University Institutional Animal Care and Use Committee. Ketamine (40 mg/kg) and Xylazine (10 mg/kg) were used for anesthesia. After exposing and opening the auditory bulla, a silver wire electrode was placed in the round window niche. The electrode was used to continuously track the amplitude of the cochlear potentials evoked by a pair of tones at 18 and 18.9 kHz. Whenever the amplitude declined, surgery was temporarily halted to allow recovery. An opening in the basal cochlear turn was used to expose the basilar membrane, which was visualized using a ×20 objective lens with numerical aperture 0.4 (Mitutoyo Inc, Takatsu-ku, Japan). Sound-evoked basilar membrane vibration was measured by a laser velocimeter (OFV-1000, Polytec Gmbh, Waldbronn, Germany) using 10-μm gold-coated glass beads as reflectors[46].

The noise level of in vivo interferometric recordings increases at low frequencies, which can lead to problems detecting basilar membrane responses at $f_e$. To ensure an adequate low-frequency signal-to-noise ratio, 100-ms stimuli were presented either 60 or 120 times, depending on the stimulus level, and the responses averaged in the time domain. Also, the data acquisition system automatically rejected records that were influenced by the breathing movements of the deeply anesthetized animal. To further reduce noise, the animal's head was firmly attached to a custom head holder and the auditory bulla anchored by a stiff metal rod to the optical table where the experiments were performed. The measures taken to reduce low-frequency noise also stabilized the animal's head during microelectrode recordings.

Tuning curves from the basilar membrane were recorded using a lock-in amplifier (SR830, Stanford Research Systems, Sunnyvale, CA), while responses to the three-tone stimuli were sampled by a 24-bit data acquisition system (PCI-4461, National Instruments, Austin, TX), which also generated the stimuli. Both systems were controlled by custom Labview software.

Following the recording of basilar membrane motion, a glass microelectrode with approximately 1-μm tip diameter was advanced toward the organ of Corti using a motorized micromanipulator. When advancing the microelectrode through the fluid in scala tympani, a 17 kHz-tone at 70 dB SPL was continuously played through the loudspeaker and the amplified electrode output fed to a lock-in amplifier and a DC voltmeter. Penetration of the basilar membrane was evident by a transient negative potential, caused by the resting membrane potential of cells on the basilar membrane. As the electrode was further advanced, the transient negative potential was followed by a large increase in the response to the 17-kHz tone, signifying placement of the electrode tip in the fluid spaces around the outer hair cells.

Identical stimulation and averaging parameters were used for recording basilar membrane motion and organ of Corti electrical potentials.

After the microelectrode recordings, basilar membrane vibration measurements were repeated using identical acquisition settings.

**Electrode calibration**. Due to the thin wall and the impedance of the tip, a glass microelectrode behaves as a first-order lowpass filter that attenuates high-frequency signals (typical cutoff frequencies ranged between 1 and 5 kHz). To correct for this effect, we measured the frequency response of each electrode while still positioned inside the organ of Corti, using the procedures described by Baden-Kristensen and Weiss[47], (see also ref.[23,24]). The calibration data were acquired using the SR830 lock-in amplifier and used to correct the microelectrode data for the effects of the electrode filter. This correction was performed in the frequency

domain, and time domain signals (i.e., Fig. 3b, d) were generated through the inverse Fourier transform.

### Optical coherence tomography (OCT).
To probe the internal vibrations of the organ of Corti, we used a Thorlabs Telesto spectral domain OCT system with 3.4 µm axial resolution. In this system, the 1300-nm light from a superluminescent diode was projected through a custom microscope onto the organ of Corti through the intact round window membrane. The round window membrane was accessed by making a small opening in the auditory bulla of deeply anesthetized guinea pigs, using the surgical approach of Lukashkin et al.[25]. This surgical approach ensured minimal trauma and meant that compound action potential thresholds were usually preserved (preparations with threshold elevation of more than 10 dB were discarded). The best frequency of the recording location was ~30 kHz.

In the OCT system, the back-reflected light from the organ of Corti is combined with a reference beam on a sensitive optical spectrometer. Since high-frequency optical signals emanate from deeper structures than low-frequency ones, Fourier transformation was used to reconstruct the depth-dependent interference pattern from the organ of Corti. By examining the phase of successive spectra, information about the displacement of the cochlear structures was obtained[48,49]. The reflectivity of the tissue determined the noise floor (0.05–0.1 nm in good preparations). Following the death of the animal, tissue reflectivity declined, resulting in an inability to accurately measure postmortem vibrations.

The OCT system was controlled by custom Labview software that acquired 10,000 optical spectra at each position using a sampling rate of 147 kHz. Spectra were stored on disk for further off-line processing. A clock signal derived from the OCT system was used to synchronize stimulus generation with the acquisition of optical spectra. Vibration signals were averaged 400 times, and vibration data were acquired at 4–6 positions across the radial extent of the organ of Corti.

To enable the use of higher stimulus levels, one speaker generated tones 1 and 3, while the phase-varying center tone was produced by a second speaker. Both speakers were mounted in a speculum tightly fitted to the ear canal. Stimuli were presented starting at the lowest level and progressing toward higher levels.

### Stimulus generation and data acquisition.
While the three-tone stimulus that we used is no speech signal, it does allow rigorous testing of the mechanisms used for detecting the envelope, which is known to be important for understanding speech. The three-tone stimuli were 100 ms long with 5-ms rise and fall time. The frequency separation between the three tones was usually 500 Hz, except where otherwise noted. The stimuli were presented to the animal through a single loudspeaker driven by a custom power amplifier, except for the OCT recordings, where two speakers were used. Recordings of the sound pressure within the speculum confirmed that the output of the loudspeaker contained no component at the frequency of the envelope variations. Responses were sampled and stimuli generated with a 24-bit data acquisition system (PCI-4461, National Instruments, Austin, TX) controlled by custom Labview software.

### Round window recordings.
The round window recordings shown in Figs. 4, 5 were made by making a small opening in the animal's bulla and placing the tip of a Teflon-insulated silver wire directly in the round window niche. A chlorided ground wire was placed in the neck muscles, and a differential amplifier used for recording the responses to the three-tone stimulus, and to acquire compound action potential audiograms in response to single tones. Only animals with a normal initial audiogram were used for these experiments.

### Whole-cell recording from rat inner hair cells (IHCs).
Rat cochleae aged P10-P12 were dissected and the organ of Corti removed and placed into a recording chamber[50,51]. Borosilicate patch electrodes with 2.5–4 MΩ resistance were used to record from mid-apical IHCs. Data were collected using an Axopatch 200b amplifier and digitized with an A/D board controlled by JClamp software (Scisoftco). Mechanical stimulation was accomplished using a glass probe shaped to that of the IHC bundle and attached to a piezo-electric stack. The voltage command to the piezo-electric stack, lowpass filtered at 10 kHz with an 8 pole Bessel filter (Cygnus technology), was set to produce mechanical stimuli resulting in 20 to 80% activation of the mechanoelectrical transducer current. For several experiments a fluid jet was used to mechanically-stimulate the hair bundles. In this case, thin-walled glass was pulled to a tip diameter of ~7 µm, filled with external solution and placed in front of a piezo disc that was driven via the JClamp software. Stimuli were lowpass filtered at 1 kHz in this case. Data were analyzed using Origin (Microcal) or MATLAB. For data to be included the leak current needed to be less than 50 pA, the series resistance less than 10 MΩ and the mechanoelectric transducer currents greater than 600 pA when the hair cell was voltage-clamped at −84 mV. External solutions contained (in mM) 135 NaCl, 2 KCl, 2 CaCl₂, 0.5 MgCl₂, 10, 4-(2-hydroxyethyl)-1-piperazineethanesulfonic acid (HEPES), 2 pyruvate, 2 ascorbate, 6 glucose, and 2 creatine, pH was balanced to 7.4 and osmolality was 305–310 mOsm l$^{-1}$. The internal solution contained (in mM) 125 KCl, 1 ethylene glycol-bis(β-aminoethyl ether)-N,N,N′,N′-tetraacetic acid (EGTA), 10 HEPES, 3 Adenosine triphosphate (ATP), 5 Creatine Phosphate, 3 MgCl₂, 2 pyruvate, pH balanced to 7.2, and osmolality maintained 285–295 mosm l$^{-1}$.

### Statistics.
Linear mixed models were used to evaluate the effect of center-tone phase on the log-transformed amplitude of basilar membrane movement or organ of Corti potentials. The model contained a preparation-specific random intercept. To model the shape seen in Fig. 1b, the fixed effect was the absolute value of the cosine of the center-tone phase. For the data shown in Fig. 1i, a permutation test[52] was used to confirm that data points were statistically different from the system noise level for each patient. The only exception was the data point for the 90° center-tone phase for subject 2, which was not different from the noise.

### Code availability.
The computer code for data analysis and acquisition are available from the corresponding authors upon reasonable request.

## Data availability
The datasets generated during the current study are available from the corresponding authors upon reasonable request.

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

# ARTICLE

21. Avan, P., Büki, B. & Petit, C. Auditory distortions: origins and functions. *Physiol. Rev.* **93**, 1563–1619 (2013).
22. Lukashkin, A. & Russell, I. J. Dependence of the DPOAE amplitude pattern on acoustical biasing of the cochlear partition. *Hear. Res.* **203**, 45–53 (2005).
23. Zheng, J. et al. Persistence of past stimulations: storing sounds within the inner ear. *Biophys. J.* **100**, 1627–1634 (2011).
24. Fridberger, A. et al. Organ of Corti potentials and the motion of the basilar membrane. *J. Neurosci.* **24**, 10057–10063 (2004).
25. Lukashkin, A., Bashtanov, M. E. & Russell, I. J. A self-mixing laser-diode interferometer for measuring basilar membrane vibrations without opening the cochlea. *J. Neurosci. Meth.* **148**, 122–129 (2005).
26. He, W., Porsov, E., Kemp, D., Nuttall, A. L. & Ren, T. The group delay and suppression pattern of the cochlear microphonic potential recorded at the round window. *PLoS ONE* **7**, e34356 (2012).
27. Fettiplace, R. & Kim, K. X. The physiology of mechanoelectrical transduction channels in hearing. *Physiol. Rev.* **94**, 951–986 (2014).
28. Johnson, S., Beurg, M., Marcotti, W. & Fettiplace, R. Prestin-driven cochlear amplification is not limited by the outer hair cell membrane time constant. *Neuron* **70**, 1143–1154 (2011).
29. Jacob, S., Pienkowski, M. & Fridberger, A. The endocochlear potential alters cochlear micromechanics. *Biophys. J.* **100**, 2586–2594 (2011).
30. Ramamoorthy, S. et al. Filtering of acoustic signals within the hearing organ. *J. Neurosci.* **34**, 9051–9058 (2014).
31. Verpy, E. et al. Stereocilin-deficient mice reveal the origin of cochlear waveform distortions. *Nature* **456**, 255–258 (2008).
32. Hakizimana, P., Brownell, W. E., Jacob, S. & Fridberger, A. Sound-induced length changes in outer hair cell stereocilia. *Nat. Commun.* **3**, 1094 (2012).
33. Brownell, W. E., Bader, C. R., Bertrand, D. & de Ribaupierre, Y. Evoked mechanical responses of isolated cochlear outer hair cells. *Science* **227**, 194–196 (1985).
34. Chen, F. et al. A differentially amplified motion in the ear for near-threshold sound detection. *Nat. Neurosci.* **14**, 770–774 (2011).
35. Ren, T., He, W. & Barr-Gillespie, P. G. Reverse transduction measured in the living cochlea by low-coherence heterodyne interferometry. *Nat. Commun.* **7**, 10282 (2016).
36. Dalhoff, E., Turcanu, D., Zenner, H. P. & Gummer, A. W. Distortion product otoacoustic emissions measured as vibration on the eardrum of human subjects. *Proc. Natl Acad. Sci. USA* **104**, 1546–1551 (2007).
37. Kemp, D. T. Otoacoustic emissions, their origin in cochlear function, and use. *Br. Med. Bull.* **63**, 223–241 (2002).
38. Abel, C. & Kössl, M. Sensitive response to low-frequency cochlear distortion products in the auditory midbrain. *J. Neurophysiol.* **101**, 1560–1574 (2009).
39. Frank, G. & Kössl, M. The acoustic two-tone distortions 2f1-f2 and f2-f1 and their possible relation to changes in the operating point of the cochlear amplifier. *Hear. Res.* **98**, 104–105 (1996).
40. Rhode, W. S. Distortion product otoacoustic emissions and basilar membrane vibration in the 6–9 kHz region of sensitive chinchilla cochleae. *J. Acoust. Soc. Am.* **122**, 2725–2737 (2007).
41. Theunissen, E. F. & Elie, J. E. Neural processing of natural sounds. *Nat. Rev. Neurosci.* **15**, 355–366 (2014).
42. Cooper, N. P. & Rhode, W. S. Mechanical responses to two-tone distortion products in the apical and basal turns of the mammalian cochlea. *J. Neurophysiol.* **78**, 261–270 (1997).
43. Buunen, T. J. F., Festen, J. M., Bilsen, F. A. & van den Brink, G. Phase effects in a three-component signal. *J. Acoust. Soc. Am.* **55**, 297–303 (1974).
44. Oxenham, A. J., Micheyl, C. & Keebler, M. V. Can temporal fine structure represent the fundamental frequency of unresolved harmonics? *J. Acoust. Soc. Am.* **125**, 2189–2199 (2009).
45. Borra, T., Versnel, H., Kemner, C., van Opstal, A. J. & van Ee, R. Octave effect in auditory attention. *Proc. Natl Acad. Sci. USA* **110**, 15225–15230 (2013).
46. Nuttall, A. L., Dolan, D. F. & Avinash, G. Laser Doppler vibrometry of basilar membrane vibration. *Hear. Res.* **51**, 203–213 (1991).
47. Baden-Kristensen, K. & Weiss, T. F. Receptor potentials of lizard hair cells with free-standing stereocilia: responses to acoustic clicks. *J. Physiol.* **335**, 699–721 (1983).
48. Ramamoorthy, S. et al. Minimally invasive surgical method to detect sound processing in the cochlear apex by optical coherence tomography. *J. Biomed. Opt.* **21**, 25003 (2016).
49. Warren, R. L. et al. Minimal basilar membrane motion in low-frequency hearing. *Proc. Natl Acad. Sci. USA* **113**, E4304–E4310 (2016).
50. Peng, A. W., Effertz, T. & Ricci, A. J. Adaptation of mammalian auditory hair cell mechanotransduction is independent of calcium entry. *Neuron* **80**, 960–972 (2013).
51. Peng, A. W., Gnanasambandam, R., Sachs, F. & Ricci, A. J. Adaptation independent modulation of auditory hair cell mechanotransduction channel open probability implicates a role for the lipid bilayer. *J. Neurosci.* **36**, 2945–2956 (2016).
52. Maris, E. & Oostenveld, R. Nonparametric statistical testing of EEG- and MEG-data. *J. Neurosci. Meth.* **164**, 177–190 (2007).

## Acknowledgements

This study was supported by the Swedish Research Council (K2014-63X-14061-14-5 and 2017-06092), Torsten Söderberg foundation, Strategic research area for systems neuro-biology (Region Östergötland), Linköping University, and NIH-NIDCD (R01 DC-004554 to TR and R01 DC 000141 to A.L.N. and A.F.). Claus Elberling, Virum, Denmark, is acknowledged for helpful discussions, and Fredrik Elinder and Karl Grosh for comments on previous manuscript versions.

## Author contributions

A.F., A.L.N., T.R., T.L., B.C.J.M., A.J.R., S.S., S.R., J.M.H., G.B., and S.R. all contributed to the design of the project. A.F., G.B., Y.Z., T.R., S.S., J.M.H., P.C.T., T.W., A.J.R., and A.L.N. performed the research; A.F. wrote the paper, with comments and suggestions from the other authors.

## Additional information

**Competing interests:** J.M.H. is an employee of Interacoustics. The remaining authors declare no competing interests.

