## [Peer Review File · Nature Communications]

Reviewers' comments:

Reviewer # 1 (Remarks to the Author):

This is an interesting investigation of a potential mechanism for the peripheral coding of the sound envelope. The authors created a three-tone stimulus with variable envelopes but similar amplitude spectra. They then tested this stimulus in the periphery of both humans and guinea pigs (including intra-operatively in two human patients). Human recordings revealed a sensitivity to the envelope modulation. In guinea pigs, the basilar membrane did not respond to the envelope, but the Organ of Corti seemed more sensitive. This was unaffected by TTX application suggesting that blocking action potentials from the auditory nerve fibers do not contribute to this phenomenon. Patch-clamp recordings and modeling suggest that instead, this form of envelope tracking might be due to activation of mechanically-sensitive ion channels.

The experiments are provocative and technically-impressive, but the conclusion from Figure 4 seems a bit rushed. The authors don't have further means to test their hypothesis? Any idea which ion channels these might be, or what other nonlinearities contribute to envelope coding?

Do the authors have a positive control for the effects of TTX?

The examples in Figure 2b-e aren't terribly convincing, particularly differences between the waveforms in Figure 2b and 2d.

Reviewer # 2 (Remarks to the Author):

The ear is not a high-fidelity transducer and as such produces distortions. The authors used 3 different approaches (ear canal recordings in humans, mechanical and electrical recordings in the guinea pig in vivo, and electrophysiological recordings from inner hair cells in vitro) to characterize how the ear responds to amplitude modulated (AM) tones. The envelope of an AM signal corresponds to a frequency component that is not present in the stimulus but that can emerge as a quadratic distortion. The authors characterized this distortion and found that it was mostly created by the nonlinearity associated with the mechano to electrical transduction.

The subject is certainly interesting to a large audience. However, I have some hesitation about the conclusions and the novelty of the findings.

1) General comments

The article is rather short and its understanding would benefit from lengthening. In particular, the methods should be better described as some information is missing. Why were human recordings done at rather high sound amplitude (80 dB)? In vivo recordings need to be better described: were BM data always recorded after electrical potentials? Why not recording BM first since the preparation certainly deteriorates with time and the hole made in the cochlea for the electrode induces hearing loss? How many responses were averaged in the temporal traces and spectra of Fig.2? For in vitro recordings of IHC, what was the membrane potential in Fig.4a? And more importantly what was the amplitude of the stimulation in Fig.4b-d?

2) Presentation of the results

The average traces in Fig.1 (d, g and h) should display error bars if the authors want to make the point that traces are (or not) different. Similarly, if the authors want to conclude that there are no detectable peaks in the power spectra (Figs.1 and 2), the noise floor should also be shown.

I would suggest to split Fig.3 in 2 figures (panels a-d and panels e-h).

3) Validity of the results

The authors observed that the magnitude of a distortion corresponding to the envelope's frequency was modulated by the phase of the center component of the stimulus in the case of the organ of Corti (OoC) potentials but not in the case of the basilar membrane (BM) vibration (lines 131-141):

"To further characterize this envelope-tracking signal, we systematically altered the phase of the center component of the stimulus. This had no effect on the magnitude of basilar membrane motion at 500 Hz (Fig. 2f, n=7)."

This statement is not valid. The amplitude of the 500 Hz signal of the BM was not modulated because no distortions above the noise level were observed at low frequencies and not because there was no modulation. Because of the noise, the experiment could not detect quadratic distortions. To conclude that the distortion at 500 Hz is not modulated by the phase of the stimulus, the authors would need to do additional experiments. One way to reduce the noise level without affecting the magnitude of the signal is to record for a longer time since the noise level is inversely proportional to the duration of the recording. I couldn't find the number of repetitions that were used for the in vivo recordings in the manuscript; therefore I can't attest whether it is realistic to record for longer given experimental constraints.

To sum up, the authors cannot conclude that the distortion at 500 Hz is not modulated by the phase of the stimulus because they were not able to detect this distortion. It is likely that the distortion is present but at a lower level. Also it is expected that the BM distortions at the frequency of the envelope will also be modulated by the phase of the stimulus (see next comment).

4) Interpretation of the results

The paper would generally benefit from a simple mathematical analysis. Considering a Taylor expansion of a nonlinear system, it is straightforward to demonstrate that the quadratic distortions depends on the phase of the carrier frequency of the AM signal as $\cos(\Phi)$. Taking the absolute value, the results in Fig.1 (a, f, and i), Fig.2g, Fig.3 (a and b) and Fig.4d have therefore a simple explanation. Making this point clear to the reader is important because any nonlinear system stimulated by an amplitude-modulated signal will produce distortions at the frequency of the envelope that varies with $\cos(\Phi)$. There is no specific particularity of the hearing system here.

Similar analysis will also explain the behavior of cubic distortions in Fig.3e-h (and higher order distortions like $3f_1-2f_2$) although the effect on the distortion at $2f_1-f_2$ might be smaller because it is also amplified by the cochlea. For the cubic distortions, the BM and OoC recordings behave very similarly. This point questions the difference in behaviors observed in Fig.2f-g and further suggests that the difference is due to quadratic distortions that were not detected because they were below the noise floor and not because they behave differently.

5) Conclusion of the paper

Although I found the multiscale approach quite interesting, I am wondering whether the data from human recordings are necessary to support the main conclusion which is: the distortion corresponding to the envelope is not seen as a BM vibration but is created by the mechanotransduction current. As mentioned in the previous comment, the dip in the envelope for a phase of 90° is general for any nonlinear system so is not surprising and does not seem to bring further evidence about the main conclusion.

I also have several hesitations with the conclusion of the paper. I have the feeling that the results are over-interpreted. Here are some examples:

Title: "An essential role for distortions produced by the hearing organ"

In the abstract: "Hence, far from being a by-product of sensory transduction, a distorted representation of sound is critical for perception of speech, music, and animal vocalizations."

Second paragraph of the conclusion (lines 270-272): "Here we demonstrated that the amplitudes of all

of the distortion components that we were able to record, whether high or low in frequency, depended on the shape of the envelope (Fig. 3e – h). This suggests that cochlear distortions arose because of the need to detect the envelope.”

My interpretation of the results would be quite different. I would rather speculate that the ear produces large distortions as a byproduct of nonlinear amplification. Then, the auditory system has evolved to decode the envelope of the signal for two main reasons: first, because it is a signal of lower frequency and therefore it can be tracked by neuronal units; second, because it allows extracting generic features (i.e. the envelope) that can be supported by different carrier frequencies (which would be the fundamental frequency of different individuals for example).

Finally, the present work does not test whether the envelope is important for speech perception which is the last sentence of the abstract. The main conclusion of the author is different and is the following: the signal of the envelope, which is not represented in the stimulus, can be extracted because of the distortions that mostly originate from the mechanotransduction process and not from basilar membrane movements. Regarding this conclusion, I would also make clear to the reader that the nonlinearity associated with BM vibrations is also the result of mechanotransduction nonlinearity and that the two components that the authors want to dissociate originate from the same element. To conclude, I would avoid providing the reader the impression that the present work studies the role of envelope in high order perception but rather explain how the envelope could be extracted.

6) Novelty

It seems that two main points of the paper were already known or are quite straightforward. First, in agreement with previous reports, the authors could not detect the distortion corresponding to the difference tone as a BM vibration. Second, any nonlinear system will produce a distortion at f_2-f_1 that shows a dip as a function of the phase of the stimulus, unless it is purely antisymmetric and all even exponents vanish. Then the authors should better explain what is really novel in their work.

7) Additional comments

I would write f_c-f_m , f_c , and f_c+f_m instead of f_1 , f_2 , and f_3 and explain that the authors used amplitude-modulated tones. Using f_1 , f_2 , and f_3 seems too general and the authors restricted their study to 3 frequencies stimuli that had a very specific relationship. Furthermore, the term amplitude-modulated signal will probably interest readers that are not necessarily in the hearing field.

There is some confusion between harmonics, which are multiples of a fundamental frequency, and amplitude-modulated signals. In the 3rd paragraph of the introduction, the authors discuss the relevance of a nonlinear system for detecting signals composed of a fundamental and several harmonics: “This may cause several harmonics to mechanically stimulate each inner hair cell, which would then respond preferentially at the peaks that result from the interactions among the harmonics. As a result, frequency components corresponding to the envelope would appear in the auditory nerve spike pattern” First, there is no envelope of such signal. Second, there is no clear benefit of creating distortions because the interaction between the different harmonics can combine to create the fundamental but it is already present in the signal.

Reviewer # 3 (Remarks to the Author):

In Nuttall et al 2017, the authors use a tremendous breadth of experiments to determine the source of distortions that contribute to the auditory system perception of complex sound envelopes that give rise to functions such as encoding of speech sounds. It is known that cells in the brainstem can respond to sound envelope stimuli, but the mechanism that generates information about the sound envelope in the auditory system is unknown. Here, the authors use elegant experiments to clearly test changes in sound envelope in isolation from other acoustic parameters, and determine that the mechanotransduction channels of hair cells have differential responses to different sound envelopes

that could drive central auditory system responses. This distortion is not a byproduct of sensory encoding, but instead a signal that is used by the auditory system to perceive stimuli such as speech.

Overall, the research presented here is clearly analyzed and presented, and appropriate controls are used. This is an excellent collaboration of authors with broad experimental backgrounds and expertise, wrapped into a thorough and focused package. The text is occasionally dense, and the data descriptions a bit vague, as described below. The authors provide convincing evidence, although additional cochlear mechanisms contributing to sound envelope detection are not ruled out, as the authors do appropriately mention. To more completely prove that IHC MET channels are responsible for the distortion that gives rise to envelope encoding, the authors would need to find a way to block only this distortion. Methods exist that would block the MET channel function entirely, but I can't think of any that would remove the distortion in isolation. Detailed comments below primarily address readability and interpretation.

Major points

1. Throughout, there is an emphasis on the 0 degree phase-shifted stimuli (blue) yielding an enhanced response, indicating detection of the sound envelope. This is indeed a strong and significant response for the various metrics, in some measures close to the system response to primary frequencies (eg figure 2e, 4c). However, the wording in some places downplays responses to the 90 degree shifted stimuli (red) to the point of inadvertently suggesting that the system is not responding to the envelope of the 90 degree shifted stimulus. The system is responding (eg. apparent in figures 2e, 3c-d, 4f), this smaller and smoother envelope is still encoded, but the 0 degree response is differentially enhanced by the distortion added by the IHC MET channels. The difference between the responses is important, not the presence or lack of response. Please adjust language throughout to acknowledge detection of the envelope stimulus for the 90 degree shifted data. Then, address more clearly the differences between the 0 degree and 90 degree shifted responses, and how the particular distortion added by the IHC MET channel currents doesn't occur at 90 degree phase shifts, but is strong at 0 degree phase shift.
2. Use actual data points and comparisons throughout, instead of relying on qualitative terms such as large or small, or "fell sharply" (eg results text figure 1e, 2c, 2e, etc).
- 3.

Minor points

1. Address how the envelope variations used as test stimuli relate to human speech or other behaviorally relevant sounds.
2. Typos: line 36, change "now" to "not"
3. The wording of the sound stimulus for humans is confusing when comparing results text, figures, and methods. The methods clearly state tone frequencies of 3750, 4062.5, and 4375 Hz (~300 Hz difference). However, results text and figure 1 legend refer to tones separated by 500 Hz. Was this used to simplify examples and figure making, or were tones with smaller separation chosen to give large peak to peak envelope frequencies at 500 Hz?
 - a. Describe envelope frequencies expected with real stimuli used in experiments, and relate to figure 1a-1b, compared to 1d and 1e. Is referring to tone frequency separation of 500 Hz a typo for figure 1? Tone frequencies are listed more clearly for figures 2 and 3 in legend.
 - b. Figure 1f figure legend "at the frequency of the envelope variations" – again, what is the frequency of the envelope variations for real test stimuli? Ok if actually 500 Hz, if not, state directly.
 - c. Figure 2 clearly uses a tone frequency separation of 500 Hz, and also has envelope frequency separations at 500 Hz. Clarify this in relation to the tone and envelope frequencies used for human experiments in figure 1.
4. Zoom in on data in figure 1d (could use -1 to +1 scale)
5. Test frequencies used are described in different locations. For figure 1, they are listed in the

methods. For figure 2 and 3, they are listed in the figure legend. Make this consistent.

6. Figure 2c and 2e – describe or indicate the noise floor

7. Figure 2c and 2e – show zoom of ~500-1500 Hz responses, with finer grained frequency axis labels

8. Figure 3 throughout – blue and red are used everywhere else to distinguish between 90 and 0 degree phase shifted stimuli. Use a different color scheme here.

9. Figures 3e-3g show that high frequency distortion products are also affected by envelope frequency, both in basilar membrane responses and organ of Corti responses. Does the auditory system also use these responses to encode envelope variations? If so, does this also argue for a role of the BM in addition to the OoC in envelope detection?

10. Figure 4 results text and figure 4b legend – the IHC are not stimulated by the three-tone stimulus, there is no sound tone involved. Instead, they are stimulated by the stiff probe (or fluid jet) replicating the frequency patterns of the three-tone stimulus. Address this throughout, and provide more details of the stimuli used.

11. Figure 4b,c legend typo line 5: “pealk”

12. Address whether it is appropriate to use the same stimulus frequencies as the original sound stimuli. Do middle and inner ear apparatus impose filtering, especially at the level of basilar membrane vibration and cilia shearing against the tectorial membrane?

13. Why was a frequency separation of 100 Hz used for the IHC recording experiments, while a 500 Hz separation was used for other experiments? Then, why does the model switch back to consideration of a 500 Hz repetition? Either repeat IHC recordings as in figure 4b-d with a 500 Hz separation, or thoroughly address the rationale for the different stimuli compared to figures 1-3.

14. Methods for IHC recordings – describe more clearly when glass probe vs fluid jet stimulation is used, or rationale for using either type.

Response to the reviewers

This is an interesting investigation of a potential mechanism for the peripheral coding of the sound envelope. The authors created a three-tone stimulus with variable envelopes but similar amplitude spectra. They then tested this stimulus in the periphery of both humans and guinea pigs (including intra-operatively in two human patients). Human recordings revealed a sensitivity to the envelope modulation. In guinea pigs, the basilar membrane did not respond to the envelope, but the Organ of Corti seemed more sensitive. This was unaffected by TTX application suggesting that blocking action potentials from the auditory nerve fibers do not contribute to this phenomenon. Patch-clamp recordings and modeling suggest that instead, this form of envelope tracking might be due to activation of mechanically-sensitive ion channels.

We thank the reviewer for these positive comments.

The experiments are provocative and technically-impressive, but the conclusion from Figure 4 seems a bit rushed. The authors don't have further means to test their hypothesis?

The figure shows recordings of mechanically evoked currents from inner hair cells. The recordings demonstrate that the envelope is encoded in the electrical potentials of the cell, and that this envelope encoding is caused by the nonlinear activation curve of the MET channels.

Voltage-clamp was chosen as a means of directly assessing the role of mechanotransduction in generating nonlinearities, with the idea that if it did not happen here further evaluation of mechanical stimulation would not be warranted. Voltage-clamp allows for isolation of the mechanotransduction response and probing of the response across frequencies and stimulus intensities. We used stimulus intensities that varied between 20-80% of maximal activation using fluid jet as well as stiff probe stimulators to ensure common outcomes independent of stimulus mode. We have elaborated the types of experiments performed in the text which originally was truncated to meet space requirements.

Current-clamp recordings could allow us to assess contributions from voltage-gated channels, but these recordings would need to be performed in older animals and preferably at body temperature, both of which makes them exceedingly difficult. Additionally, recordings in endolymph-like solutions could be valuable, but to date we have not been able to gain accurate separation of perilymph and endolymph while patch-clamping the hair cells. Thus, although receptor potential measurements would be ideal, they are not yet technically possible at the level of rigor needed for publication. However, the voltage-clamp data demonstrate the feasibility of the mechanism and the likelihood of the mechanotransduction properties translating to receptor potentials is very high.

Any idea which ion channels these might be, or what other nonlinearities contribute to envelope coding?

We think that nonlinearity in the MET channel is the main mechanism, but it is possible that active bundle mechanics contribute (Barral and Martin, PNAS 2012). Some experiments show a higher apparent stiffness when bundles are subjected to large deflections (Flock and Strelioff, Nature 1984), but it is not known whether this phenomenon is relevant to the small deflections observed during sound stimulation. Finally, Verpy et al (Nature 2008) suggested

that the protein stereocilin could contribute to asymmetric bundle mechanics. These papers are cited in the revised version. If any of these mechanisms contribute, they would mainly affect high-frequency distortion products (which we show to be significantly affected by the shape of the envelope).

Stiff probe results compared to fluid jet results may be a way to test intrinsic bundle mechanical properties and given the similarity in results we would argue that mechanics is probably not a major contributor; however micromechanical contributions can occur even with a stiff probe and so it is feasible that mechanics associated with gating or fast adaptation could still be involved.

Voltage-gated ion channels described in cochlear hair cells contribute to both receptor potentials and the resting membrane potential. In mature hair cells these conductances appear to be important for speeding up the membrane time constant and linearizing the receptor potential, largely ensuring that the signal from mechanotransduction is accurately reconstructed and not so much involved in shaping or filtering the response (as in lower frequency end organs).

Do the authors have a positive control for the effects of TTX?

We measured compound action potential responses to high-frequency tone bursts and found them nearly abolished after TTX. This information was added to the results section.

The examples in Figure 2b-e aren't terribly convincing, particularly differences between the waveforms in Figure 2b and 2d.

The difference between the 0 and 90° phases is easier to see in the spectra, which reveal pronounced low-frequency distortion in Fig. 2e as well as high-frequency distortions in both Fig. 2c and e. However, we thought it prudent to show both time- and frequency-domain data. If the reviewer finds it essential, we can eliminate the time-domain records.

Reviewer #2

The ear is not a high-fidelity transducer and as such produces distortions. The authors used 3 different approaches (ear canal recordings in humans, mechanical and electrical recordings in the guinea pig in vivo, and electrophysiological recordings from inner hair cells in vitro) to characterize how the ear responds to amplitude modulated (AM) tones. The envelope of an AM signal corresponds to a frequency component that is not present in the stimulus but that can emerge as a quadratic distortion. The authors characterized this distortion and found that it was mostly created by the nonlinearity associated with the mechano to electrical transduction. The subject is certainly interesting to a large audience. We thank the reviewer for these positive comments, but we would like to point out that the stimulus we used is not an amplitude-modulated sound in the usual meaning of the term, as described in more detail in the revised manuscript as well as in the text below.

However, I have some hesitation about the conclusions and the novelty of the findings.

We hope that the changes detailed below, and the new experiments, will be sufficient to convince the reviewer about the validity of the conclusions.

As regards novelty, we would point out that all previous studies examined parts of the system in isolation, some attributing envelope coding to the characteristics of auditory nerve fibers, others to synaptic properties, and yet others to different mechanisms within the hair cells. Our study is the first to systematically evaluate these mechanisms – and it is important to do so, since envelope coding is critical for communication.

The new optical coherence tomography (OCT) experiments demonstrate a previously unknown vibration pattern within the organ of Corti, and we are grateful to the reviewer for making us perform these experiments. Nothing similar is available in the literature, so these experiments on their own are clearly novel.

We also found that all distortion products were affected by the envelope shape. To our knowledge, this possibility has never previously been considered. It is an important effect that brings a new idea to the ongoing debate about the origins and functions of distortion products.

Finally, the stimulus we used will be useful to others since large changes to the envelope can be made without altering the frequency spectrum of the stimulus. This considerably facilitates interpretation of the results, and it is impossible to achieve when using conventional amplitude modulation.

The article is rather short and its understanding would benefit from lengthening. In particular, the methods should be better described as some information is missing.

The methods section was expanded to include additional details. We also include new OCT results, more details about the mechanotransduction responses, and additional discussion.

Why were human recordings done at rather high sound amplitude (80 dB)?

Signals recorded from the ear canal are small, resulting in a need for extensive averaging and a long recording time. Even with the current parameters, a complete set of measurements takes approximately 1 h. Recordings from the promontory are faster, but surgical considerations make it important to limit the recording time as much as possible.

In vivo recordings need to be better described: were BM data always recorded after electrical potentials? Why not recording BM first since the preparation certainly deteriorates with time and the hole made in the cochlea for the electrode induces hearing loss?

We recorded BM data both before and after electrode penetration. Since electrode penetration can result in some hearing loss, we thought it prudent to present the BM data acquired after the electrode penetration. As regards envelope-following frequency components in the BM vibration, there was no difference between recordings made before and after the electrode penetration. In neither case could an envelope-following component be detected.

How many responses were averaged in the temporal traces and spectra of Fig.2?

The number of averages varied depending on the stimulus level. 120 averages were used at 44 dB SPL and 60 averages at 64 dB SPL. The stimulus duration was 100 ms in all cases.

For in vitro recordings of IHC, what was the membrane potential in Fig.4a? And more importantly what was the amplitude of the stimulation in Fig.4b-d?

The holding potential used for these experiments was -84 mV, selected to exclude changes in voltage-gated conductances and also to provide a driving force for current flow similar to that expected *in vivo* (more hyperpolarized would be better but cells are hard to sustain below -100 mV). Stimulation amplitudes were set so as to evoke 20-80% of maximal mechanotransduction currents. The absolute amplitude depended on mode of stimulation as well as on dimensions of stiff probe used. With fluid jet stimulation it was difficult to monitor absolute hair bundle motions using the complex sinusoidal waveforms but in general the motions were not detectable when observing the bundle on screen (meaning less than ~300nm).

The average traces in Fig.1 (d, g and h) should display error bars if the authors want to make the point that traces are (or not) different.

Figures 1d and e were intended to show example recordings from single subjects. As such, they represent the average of 50 000 stimulus presentations. However, the variability in the amplitudes across subjects and stimulus conditions is apparent from Fig. 1f, where individual data points are shown (dots) together with the mean and the standard error. The differences were statistically significant.

Similarly, if the authors want to conclude that there are no detectable peaks in the power spectra (Figs.1 and 2), the noise floor should also be shown.

The noise floor near the frequency of the envelope peak should be apparent in figure 1e and i. In figure 2, we changed the axis limits in panels c and e to better show the noise floor.

I would suggest to split Fig.3 in 2 figures (panels a-d and panels e-h).

Done.

The authors observed that the magnitude of a distortion corresponding to the envelope's frequency was modulated by the phase of the center component of the stimulus in the case of the organ of Corti (OoC) potentials but not in the case of the basilar membrane (BM) vibration (lines 131-141): "To further characterize this envelope-tracking signal, we systematically altered the phase of the center component of the stimulus. This had no effect on the magnitude of basilar membrane motion at 500 Hz (Fig. 2f, n=7)." This statement is not valid. The amplitude of the 500 Hz signal of the BM was not modulated because no distortions above the noise level were observed at low frequencies and not because there was no modulation. Because of the noise, the experiment could not detect quadratic distortions. To conclude that the distortion at 500 Hz is not modulated by the phase of the stimulus, the authors would need to do additional experiments. One way to reduce the noise level without affecting the magnitude of the signal is to record for a longer time since the noise level is inversely proportional to the duration of the recording. I couldn't find the number of repetitions that were used for the in vivo recordings in the manuscript; therefore I can't attest whether it is realistic to record for longer given experimental constraints.

We appreciate these comments and in response, the results and the methods sections were rewritten and new experimental data included.

We averaged either 120 or 60 individual responses, depending on the stimulus level, using stimuli that were 100 ms long. The recordings were done using a range of stimulus frequencies covering the region around the best frequency of the recording location. At each frequency, 7 different recordings were performed to cover the range of center-tone phases. Hence, all the data presented in the paper are extensively averaged, and the methods section now provides the details requested by the reviewer.

To further reduce low-frequency noise, the data acquisition system automatically rejected records influenced by animal motion. Additionally, the anesthetized animal's head was firmly attached to a metal head holder and a stiff metal rod glued to the auditory bulla. The metal rod was screwed into the optical table to rigidly fix the bulla. This is now all described in the text.

The extensive averaging and artefact rejection substantially reduced low-frequency noise, but also led to lengthy data collection. The initial recording of BM motions was usually completed in about 1 h. After this, the electrode was moved into position and electrical potentials recorded (total time about 1.5 h for positioning the electrode and performing recordings and electrode calibrations). Finally, the electrode was withdrawn and the BM motion again recorded. In most experiments, this entire data collection took around 4 hours. To substantially increase the signal-to-noise ratio, the number of averages would need to be at least doubled. Unfortunately, such a long data collection would lead to problems maintaining hearing sensitivity during the experiment. The revised version has a better description of these procedures.

The reviewer's concern about the signal-to-noise ratio is important, and we therefore performed new experiments using OCT. In sensitive animals, an envelope-following component was evident in the vibrations of the reticular lamina, but only at stimulus levels at and above 74 dB SPL. This frequency component was not detected in the basilar membrane's motion at any stimulus level, despite noise floors that were near or in some cases below previously described displacements measured at auditory threshold (the new figure 3 presents an example where the basilar membrane's noise floor was around 0.06 nm; Lukashkin et al (Hearing Res 2005) showed displacement of 0.09 nm at 10 dB SPL for this region of the cochlea). Note that OCT records the vibrations of the basilar membrane and the reticular lamina simultaneously, so time-dependent changes do not explain the difference between the structures. These findings suggest a complicated processing of envelope signals within the organ of Corti.

In contrast, the microelectrode as well as the round window recordings show envelope-following components at low stimulus levels.

In summary, these data establish that the stimulus envelope is not apparent in the vibrations of the basilar membrane, although it is present at the reticular lamina at moderate and high stimulus levels, and clearly present in the cochlear potentials at low levels.

To sum up, the authors cannot conclude that the distortion at 500 Hz is not modulated by the phase of the stimulus because they were not able to detect this distortion. It is likely that the distortion is present but at a lower level.

As explained above, the new OCT experiments show that distortion can be detected at the reticular lamina when the stimulus level is 74 dB SPL or above. No similar component could be observed at the basilar membrane, despite noise floors corresponding to the limit of auditory sensitivity. In contrast, electrical distortion was easily detected at 44 dB SPL.

In addition, even at 74 dB SPL, the mechanical distortion at the reticular lamina was 15 - 20 dB below the level of the primaries, whereas the microelectrode recordings showed electrical distortion 6 dB below the level of the primaries. Hence, mechanical distortion at the envelope frequency is a high-level phenomenon, but electrical distortion is not.

We would point out that the OCT recordings add substantially to the paper and that no similar measurement has previously been reported.

Also it is expected that the BM distortions at the frequency of the envelope will also be modulated by the phase of the stimulus (see next comment).

As regards modulation of a putative BM component, we agree that it should be modulated by the envelope, had it been present. But we were unable to find any evidence for it, despite extensive averaging of BM motions in highly sensitive ears, using both laser interferometry and OCT.

The paper would generally benefit from a simple mathematical analysis. Considering a Taylor expansion of a nonlinear system, it is straightforward to demonstrate that the quadratic distortions depends on the phase of the carrier frequency of the AM signal as $\cos(\Phi)$. Taking the absolute value, the results in Fig.1 (a, f, and i), Fig.2g, Fig.3 (a and b) and Fig.4d have therefore a simple explanation. Making this point clear to the reader is important because any nonlinear system stimulated by an amplitude-modulated signal will produce distortions at the frequency of the envelope that varies with $\cos(\Phi)$. There is no specific particularity of the hearing system here.

We agree that additional mathematical analysis can be beneficial for understanding the results. As suggested by the reviewer, the effects of the distortion in the auditory system can be written as a Taylor series containing linear, quadratic, and cubic terms with decreasing amplitudes. If a three-tone stimulus is passed through such a nonlinearity, the quadratic term generates a component at the frequency of the envelope variations, the amplitude of which depends on the phase of the center tone of the stimulus. This reasoning was added to the results section.

While this analysis is illuminating, it does not address other contributing mechanisms proposed in the literature. These potential mechanisms are however addressed by the other experiments reported in the manuscript.

We also note that the Taylor expansion predicts frequency components not observed experimentally, including components near the second harmonic of F1.

Similar analysis will also explain the behavior of cubic distortions in Fig.3e-h (and higher order distortions like $3f_1-2f_2$) although the effect on the distortion at $2f_1-f_2$ might be smaller because it is also amplified by the cochlea. For the cubic distortions, the BM and OoC recordings behave very similarly. This point questions the difference in behaviors

observed in Fig.2f-g and further suggests that the difference is due to quadratic distortions that were not detected because they were below the noise floor and not because they behave differently.

As mentioned above, several measures were taken to minimize low-frequency noise. Moreover, we performed additional experiments using OCT. These experiments revealed a quadratic component at the reticular lamina at moderate and high stimulus levels. We could not detect such a component at the basilar membrane, and it was not detectable from either structure at lower sound pressure levels, despite noise floors that were sometimes better than 0.1 nm.

To our knowledge, envelope effects on high-frequency distortions have never before been suggested in the literature, and no previous measurement suggested their presence. Even if such an effect is predicted by the Taylor expansion, this model also predicts distortion components that cannot be observed experimentally, which limits its explanatory power.

Also, when we used a simple but physiologically relevant model based on the MET channel nonlinearities, an envelope-following component was always generated – but we could not reproduce the envelope effects on the high-frequency distortions. Hence, the situation is more complicated than the reviewer suggests.

Although I found the multiscale approach quite interesting, I am wondering whether the data from human recordings are necessary to support the main conclusion which is: the distortion corresponding to the envelope is not seen as a BM vibration but is created by the mechanotransduction current. As mentioned in the previous comment, the dip in the envelope for a phase of 90° is general for any nonlinear system so is not surprising and does not seem to bring further evidence about the main conclusion.

As mentioned above, the model suggested by the reviewer does predict some of the findings, but it also predicts things that are not observed experimentally.

Species differences are a fact of life, and things often do not work as one would expect when moving from one species to a different one. How can we know that human auditory system works in the same way unless someone shows it? We note that the number of published human recordings is limited, and none of them looked an envelope coding.

I also have several hesitations with the conclusion of the paper. I have the feeling that the results are over-interpreted. Here are some examples:

Title: “An essential role for distortions produced by the hearing organ”

In the abstract: “Hence, far from being a by-product of sensory transduction, a distorted representation of sound is critical for perception of speech, music, and animal vocalizations.”

To avoid over-interpreting our results, we changed the title and the abstract along the lines suggested by the reviewer, and there are also substantial changes to the results section.

Second paragraph of the conclusion (lines 270-272): “Here we demonstrated that the amplitudes of all of the distortion components that we were able to record, whether high or low in frequency, depended on the shape of the envelope (Fig. 3e – h). This suggests that cochlear distortions arose because of the need to detect the envelope.”

My interpretation of the results would be quite different. I would rather speculate that the ear produces large distortions as a byproduct of nonlinear amplification. Then, the auditory system has evolved to decode the envelope of the signal for two main reasons: first, because it is a signal of lower frequency and therefore it can be tracked by neuronal units; second, because it allows extracting generic features (i.e. the envelope) that can be supported by different carrier frequencies (which would be the fundamental frequency of different individuals for example).

Finally, the present work does not test whether the envelope is important for speech perception which is the last sentence of the abstract. The main conclusion of the author is different and is the following: the signal of the envelope, which is not represented in the stimulus, can be extracted because of the distortions that mostly originate from the mechanotransduction process and not from basilar membrane movements. Regarding this conclusion, I would also make clear to the reader that the nonlinearity associated with BM vibrations is also the result of mechanotransduction nonlinearity and that the two components that the authors want to dissociate originate from the same element. To conclude, I would avoid providing the reader the impression that the present work studies the role of envelope in high order perception but rather explain how the envelope could be extracted.

We rewrote the abstract, introduction, and discussion to focus on how the envelope is extracted by the cochlea, which is a very important problem in its own right.

It seems that two main points of the paper were already known or are quite straightforward. First, in agreement with previous reports, the authors could not detect the distortion corresponding to the difference tone as a BM vibration. Second, any nonlinear system will produce a distortion at f_2-f_1 that shows a dip as a function of the phase of the stimulus, unless it is purely antisymmetric and all even exponents vanish. Then the authors should better explain what is really novel in their work.

As explained in the revised abstract, there are several theories attempting to explain how the envelope is extracted. References cited in the introduction include at least three different possible mechanisms, but there has been no previous attempt to determine which one of these possible mechanisms may be the most important one. Our recordings show that envelope extraction by auditory nerve dendrites (as proposed by Carney et al, and others) is not a dominant mechanism, since envelope coding remains when auditory nerve activity is blocked. Similarly, the theory proposed by Sayles and Winter (Neuron 2008) is not supported by our data.

This leaves the distortion theory as the only one with strong experimental support. The reviewer seems to consider this idea obvious, but how can we know this unless the idea is experimentally tested against the other theories?

Furthermore, OCT reveals interesting processing of envelope signals within the organ of Corti that was not possible to assess with previous techniques.

I would write f_c-f_m , f_c , and f_c+f_m instead of f_1 , f_2 , and f_3 and explain that the authors used amplitude-modulated tones. Using f_1 , f_2 , and f_3 seems too general and the authors restricted their study to 3 frequencies stimuli that had a very specific relationship.

Furthermore, the term amplitude-modulated signal will probably interest readers that are not necessarily in the hearing field.

We appreciate this comment, but the problem is that the stimuli we used are not amplitude-modulated tones in the conventional sense. An amplitude-modulated signal is defined by

$$y(t) = [1 + M \cos(2\pi f_m t + \varphi)] \times A \sin(2\pi f_c t)$$

where f_m is the modulation frequency, f_c the carrier frequency, and M the modulation depth. When using such sounds, it is impossible to change the envelope without changing the magnitude of the primaries (the magnitude at $f_c + f_m$ and $f_c - f_m$ always changes when the modulation depth is altered). This makes conventional amplitude-modulated sounds rather difficult to use for studying the effects of the envelope on high-order distortion products, which perhaps is the reason that the effect of the envelope on distortion products has not previously been noted, studied, or considered.

In contrast, the three-tone stimulus used in our work allows large changes to the envelope without changing the amplitude of the primaries, which considerably facilitates interpretation. We made changes to the introduction and the results section to highlight this important difference.

There is some confusion between harmonics, which are multiples of a fundamental frequency, and amplitude-modulated signals. In the 3rd paragraph of the introduction, the authors discuss the relevance of a nonlinear system for detecting signals composed of a fundamental and several harmonics: “This may cause several harmonics to mechanically stimulate each inner hair cell, which would then respond preferentially at the peaks that result from the interactions among the harmonics. As a result, frequency components corresponding to the envelope would appear in the auditory nerve spike pattern” First, there is no envelope of such signal. Second, there is no clear benefit of creating distortions because the interaction between the different harmonics can combine to create the fundamental but it is already present in the signal.

We should emphasize that this is not our description, it is a summary of the arguments made by Sayles and Winter in their 2008 Neuron paper. In this paper, they suggested a mechanism for envelope detection, which is the reason the work is described and cited here.

This reviewer comment and the papers cited in the introduction illustrate why our paper is important. There are several theories about envelope detection in the literature, few attempts to evaluate them experimentally, and considerable confusion.

Reviewer #3

In Nuttall et al 2017, the authors use a tremendous breadth of experiments to determine the source of distortions that contribute to the auditory system perception of complex sound envelopes that give rise to functions such as encoding of speech sounds. It is known that cells in the brainstem can respond to sound envelope stimuli, but the mechanism that generates information about the sound envelope in the auditory system is unknown. Here, the authors use elegant experiments to clearly test changes in sound envelope in isolation from other acoustic parameters, and determine that the mechanotransduction channels of hair cells have differential responses to different sound envelopes that could drive central

auditory system responses. This distortion is not a byproduct of sensory encoding, but instead a signal that is used by the auditory system to perceive stimuli such as speech.
We thank the reviewer for these positive comments.

Overall, the research presented here is clearly analyzed and presented, and appropriate controls are used. This is an excellent collaboration of authors with broad experimental backgrounds and expertise, wrapped into a thorough and focused package. The text is occasionally dense, and the data descriptions a bit vague, as described below.

We agree that the text is sometimes too dense, and made efforts to revise it according to this and the other reviewer's suggestions.

The authors provide convincing evidence, although additional cochlear mechanisms contributing to sound envelope detection are not ruled out, as the authors do appropriately mention. To more completely prove that IHC MET channels are responsible for the distortion that gives rise to envelope encoding, the authors would need to find a way to block only this distortion. Methods exist that would block the MET channel function entirely, but I can't think of any that would remove the distortion in isolation. Detailed comments below primarily address readability and interpretation.

We agree with the reviewer that an experiment where only the distortion was blocked would be valuable, but as noted by the reviewer, no such tool currently exists.

Throughout, there is an emphasis on the 0 degree phase-shifted stimuli (blue) yielding an enhanced response, indicating detection of the sound envelope. This is indeed a strong and significant response for the various metrics, in some measures close to the system response to primary frequencies (eg figure 2e, 4c). However, the wording in some places downplays responses to the 90 degree shifted stimuli (red) to the point of inadvertently suggesting that the system is not responding to the envelope of the 90 degree shifted stimulus. The system is responding (eg. apparent in figures 2e, 3c-d, 4f), this smaller and smoother envelope is still encoded, but the 0 degree response is differentially enhanced by the distortion added by the IHC MET channels. The difference between the responses is important, not the presence or lack of response. Please adjust language throughout to acknowledge detection of the envelope stimulus for the 90 degree shifted data. Then, address more clearly the differences between the 0 degree and 90 degree shifted responses, and how the particular distortion added by the IHC MET channel currents doesn't occur at 90 degree phase shifts, but is strong at 0 degree phase shift.

We agree, and revised the text to emphasize that the 90° condition mostly means a strong reduction of the envelope response and not its complete absence.

Use actual data points and comparisons throughout, instead of relying on qualitative terms such as large or small, or "fell sharply" (eg results text figure 1e, 2c, 2e, etc).

We changed parts of the results section to avoid qualitative language.

Address how the envelope variations used as test stimuli relate to human speech or other behaviorally relevant sounds.

A brief description is included in the methods section, starting at line 484.

Typos: line 36, change "now" to "not"

Done.

The wording of the sound stimulus for humans is confusing when comparing results text, figures, and methods. The methods clearly state tone frequencies of 3750, 4062.5, and 4375 Hz (~300 Hz difference). However, results text and figure 1 legend refer to tones separated by 500 Hz. Was this used to simplify examples and figure making, or were tones with smaller separation chosen to give large peak to peak envelope frequencies at 500 Hz? The guinea pig experiments used a 500-Hz frequency separation between the primaries, while human experiments utilized a 300-Hz separation. We do not think the difference is important, but we agree this can be confusing. We therefore removed references to 500 Hz at the beginning of the results section to make it more general.

- a. **Describe envelope frequencies expected with real stimuli used in experiments, and relate to figure 1a-1b, compared to 1d and 1e. Is referring to tone frequency separation of 500 Hz a typo for figure 1? Tone frequencies are listed more clearly for figures 2 and 3 in legend.**

The legend to figure 1 was rewritten.

- b. **Figure 1f figure legend “at the frequency of the envelope variations” – again, what is the frequency of the envelope variations for real test stimuli? Ok if actually 500 Hz, if not, state directly.**

The legend now states that the frequency was 300 Hz.

- c. **Figure 2 clearly uses a tone frequency separation of 500 Hz, and also has envelope frequency separations at 500 Hz. Clarify this in relation to the tone and envelope frequencies used for human experiments in figure 1.**

As mentioned above, we rewrote the legend for figure 1 and the beginning of the results section.

Zoom in on data in figure 1d (could use -1 to +1 scale)

Done.

Reviewers' comments:

Reviewer # 1 (Remarks to the Author):

The authors have adequately addressed my criticisms (and I think the criticisms of the other reviewers). They changed the title to something a little generic though, I'd recommend including something about the nature of mechanotransduction mechanism in the title to make it more informative.

Reviewer # 2 (Remarks to the Author):

The authors provide a substantial piece of evidence demonstrating that the transduction current generated by inner hair cells produces prominent distortions that could be used by the auditory system to detect the envelop of the sound vibration. The additional experiment (optical coherence tomography) brings new argument and complements elegantly the picture. Altogether the work seems very solid and answers an important question.

Although I support publication in Nature Communications, I would like to point out several elements in the writing, presentation, and analysis of data (there is no need of additional experiments) that would need to be addressed before publication. Also the authors should take into account the comments of all reviewers more thoroughly to improve the manuscript. For example, Reviewer 3 asked whether the frequency separation of 500 Hz was a typo? In fact, after careful examination it seems that Fig. 1a shows frequencies that were used for in vivo recordings whereas the figure displays Human recordings where frequencies (and their relative separation) were different. It seems that this comment has been overlooked (see comment about Figure 1 for improvement).

Major points

1) Envelope

There are two main contributions in the Taylor expansion that explain the shape of the envelope which is the principal focus of this study. The first one at f_m has an amplitude that is modulated by $\cos(\phi)$ and the second at $2f_m$ has a constant amplitude. The authors focused solely on the frequency f_m whereas the envelope at $2f_m$ is equally important. It would considerably strengthen the paper if the authors could also show the amplitude of this distortion (at $2f_m$) as a function of ϕ in the following figures: Figs. 1f, 1i, 2f-g, 4, and 6d. The expectation is that the amplitude is fairly flat as ϕ is varied.

In every figures, the authors labelled only the peak at f_m which is misleading ("envelope peak" (Fig. 1e), "low-frequency distortions" (Fig. 2c-d), "envelope signal" (Fig. 3), "envelope peak" (Fig. 6)). The envelope in the general case is composed of the 2 terms described above, although additional terms could also contribute. To be more precise, the authors should label the peaks (f_m and $2f_m$) that are apparent in most power spectra. This is a comment that was also addressed by one reviewer who noticed that the response to the phase 90 stimulus also displayed distortions corresponding to the envelope.

2) Figures and analysis

There are some examples where it remains unclear how the data were analyzed and what has actually been plotted in the figures.

a) In particular, how data were averaged? When authors say that a 100 ms tone at 16 kHz was averaged 60 times (for Fig. 2 for example), it seems that the 100 ms tone was presented 60 times and simply averaged. But sound is a periodic signal, so why not averaging the 50 cycles of the

envelope that a 100 ms tone contains (if 500 Hz is the modulation frequency)? It would give an average over 3,000 repetitions instead of 60 which could considerably reduce the noise in the temporal traces presented in Figs. 1d, 2b, 2d, and 6b. A previous comment from one reviewer was that Fig. 2b-d aren't terribly convincing. Maybe averaging over each cycle of the envelope (at fm) could permit to see the effect on the temporal trace and not only on the power spectrum. Finally, the author could low-pass filter the signal in Fig. 2b and d (let's say at 5 kHz) and also display the filtered signal below the raw traces to represent visually the envelope. This analysis could also be done for Fig.6.

b) I agree that averaging is not going to change the power spectrum which was most probably computed over the whole 100 ms segment. But was the power spectrum calculated over the averaged temporal trace or were power spectra calculated over each single trace and then averaged?

c) Was there any window used to compute the power spectrum? It seems that some peaks are wide. If the authors have used integer numbers of cycles (which seems to be the case), all the power at the primaries and at the combination tones should be contained in a single point in the power spectrum.

d) Were the power spectra down-sampled for representation?

e) It is difficult to see ticks in almost all figures (especially the log ones).

3) Presentation of the new paradigm

As the authors argue, the use of the 3-tone stimulus is novel and enables probing the auditory nonlinearity cleverly. It might be worth making a figure with Fig. 1a and Fig. 1b only in order to present this new stimulus and discuss the expected results. The author should also present the amplitude of the 2fm envelope in Fig. 1b (see point 1). I would write the equation of the 3-tone stimulus there and not in the "Mechano-electrical transduction channels generate envelope-tracking responses" section that comes too late in the paper. This is the stimulus that was used all along the study and not only in the patch-clamp experiments. I would also include the discussion about the Taylor expansion because it is general to all approaches and gives useful intuition about the results.

4) Figure 3 and 6

It is probably important to show data on a linear frequency scale since it was represented in this way in the other figures. This is necessary to support the following statement related to Fig. 3: "High-frequency distortion products were however present on both structures." At the present time, it is impossible to tell whether there are distortions around the primaries because the scale is too small to distinguish anything. Alternatively, a zoom of the spectra around the primaries could be shown but a linear axis would be more favorable to compare data between figures.

5) Consistency

The authors should try to stay consistent between figures and show the same number of cycles of the envelope in each figure. Again, this is important to compare figures. Let's say that the authors want to show a bit more than 1 cycle at fm as they did in Fig. 1a and in Fig. 2b-d which is good to see the envelope and the primaries. In contradistinction, Fig. 1d shows about 10 cycles and Fig. 6b shows 9 cycles.

6) Patch-clamp and simulations

Why did the authors use different frequencies in patch-clamp and simulation experiments? I understand that it would not be possible to use 15 kHz in voltage-clamp recordings but why not in simulations using the frequencies used in experiments? There is no temporal component in the described model, so stimulation at 15 kHz or at 15 Hz will give the exact same result. It should also be made clear to the reader that the author considered the mechano-transduction channels as infinitely fast in their model.

Would it be also possible to show the cubic distortions in the patch-clamp experiment and in the model (the same ones as in Fig. 5)? This could be added in a supplementary figure.

Also to clarify that distortions arise for a sufficient stimulus level, it would be interesting to show (maybe in an additional supplementary figure) the relative amplitude of the envelope for low (<1nm, when there is no distortion), medium (around 10 nm, when the relative amplitude of distortion is large) and large (>100 nm, when distortions saturate) displacements X for a given X0 (let's say at X0=7 nm where distortions are the highest).

Minor points

Line 169: "but this model also predicts responses not observed experimentally, such as frequency components centered on the second harmonic of the stimulus." It is difficult to judge because the authors don't show the spectrum around these frequencies.

Line 187: "2-dB loss of auditory sensitivity" This accuracy seems quite remarkable. Is this a typo? If not, please explain in the Methods how the auditory threshold was measured and what the accuracy of the method was.

Line 442: what is the typical cutoff frequency of the electrodes?

Figure 1: The frequencies used in Fig. 1a are the one used for in vivo recordings in Guinea pigs which is quite confusing because the frequencies used for Human data are different. The author should try to be consistent and present the stimulus that was actually used in this experiment. It is also important to show temporal traces of same length in Fig.1a and 1d to compare directly the data (the stimulus and the response) and have a sense of the frequency that carries the distorted signal. After averaging (see previous point 2.a), it should be quite clear that the "center phase 0" trace oscillates at fm and the "center phase 90" oscillates at 2fm (we already see in the temporal trace that the red trace oscillates at twice the frequency of the blue trace but averaging should make it even more obvious).

Figure 3: It should be made a bit clearer that measurements at the RW are electrical. To avoid confusion, the spectrum for RW electrical potentials could be represented in another color (and maybe another color than the blue/red pair that was used for the phase of the 3-tone stimulus). It would also facilitate the understanding of the figure if the authors could show RL, BM and RW for both sound intensities (in the present manuscript, RW electrical potentials are only shown for 74 dB). If the authors don't have the data, it would help comparison between sound intensities to present RL and BM on top of each other. Finally, writing the sound intensity on the figure itself could simplify its understanding.

Reviewer # 3 (Remarks to the Author):

The authors have addressed my concerns.

Reviewer #1 (Remarks to the Author):

The authors have adequately addressed my criticisms (and I think the criticisms of the other reviewers). They changed the title to something a little generic though, I'd recommend including something about the nature of mechanotransduction mechanism in the title to make it more informative.

We changed the title to "A mechanoelectrical mechanism for detection of sound envelopes in the hearing organ"

Reviewer #2 (Remarks to the Author):

The authors provide a substantial piece of evidence demonstrating that the transduction current generated by inner hair cells produces prominent distortions that could be used by the auditory system to detect the envelop of the sound vibration. The additional experiment (optical coherence tomography) brings new argument and complements elegantly the picture. Altogether the work seems very solid and answers an important question.

We thank the reviewer for these positive comments and for taking the time to thoroughly assess our paper.

Although I support publication in Nature Communications, I would like to point out several elements in the writing, presentation, and analysis of data (there is no need of additional experiments) that would need to be addressed before publication. Also the authors should take into account the comments of all reviewers more thoroughly to improve the manuscript. For example, Reviewer 3 asked whether the frequency separation of 500 Hz was a typo? In fact, after careful examination it seems that Fig. 1a shows frequencies that were used for in vivo recordings whereas the figure displays Human recordings where frequencies (and their relative separation) were different. It seems that this comment has been overlooked (see comment about Figure 1 for improvement).

The figure legend and the text of the results section were both altered in order to more fully address the comment from reviewer 3. In addition, the figure is now separate, and we removed the scale bars to make the schematic completely generic.

Major points

1) Envelope. There are two main contributions in the Taylor expansion that explain the shape of the envelope which is the principal focus of this study. The first one at f_m has an amplitude that is modulated by $\cos(\phi)$ and the second at $2f_m$ has a constant amplitude. The authors focused solely on the frequency f_m whereas the envelope at $2f_m$ is equally important. It would considerably strengthen the paper if the authors could also show the amplitude of this distortion (at $2f_m$) as a function of ϕ in the following figures: Figs. 1f, 1i, 2f-g, 4, and 6d. The expectation is that the amplitude is fairly flat as ϕ is varied.

In response to the comments below, the previous Fig. 1a and b is now a separate figure, and the response at $2f_m$ is given in panel b. For the previous figures 1f, i, 2f-g, 4 and 6d, the inclusion of $2f_m$ data made the plots difficult to read since individual data points, means, as well as SEMs are plotted. We included $2f_m$ data in the text instead.

In every figure, the authors labelled only the peak at f_m which is misleading ("envelope peak" (Fig. 1e), "low-frequency distortions" (Fig. 2c-d), "envelope signal" (Fig. 3), "envelope peak" (Fig. 6)). The envelope in the general case is composed of the 2 terms described above, although additional terms could also contribute. To be more precise, the authors should

label the peaks (f_m and $2f_m$) that are apparent in most power spectra. This is a comment that was also addressed by one reviewer who noticed that the response to the phase 90 stimulus also displayed distortions corresponding to the envelope.

The labeling of the peaks was made consistent through all the figures. We choose to label them F_e and $2F_e$, as F_m could lead to a potential confusion with conventionally amplitude-modulated signals. This terminology was also used in the equations, which now appear at the beginning of the results section.

2) Figures and analysis

There are some examples where it remains unclear how the data were analyzed and what has actually been plotted in the figures.

a) In particular, how data were averaged? When authors say that a 100 ms tone at 16 kHz was averaged 60 times (for Fig. 2 for example), it seems that the 100 ms tone was presented 60 times and simply averaged. But sound is a periodic signal, so why not averaging the 50 cycles of the envelope that a 100 ms tone contains (if 500 Hz is the modulation frequency)? It would give an average over 3,000 repetitions instead of 60 which could considerably reduce the noise in the temporal traces presented in Figs. 1d, 2b, 2d, and 6b.

We made changes to the methods section to further clarify how the averaging was performed. The 100 ms tones were presented 60 or 120 times and the average of the responses computed. Following this, the Fourier transform was used to compute the amplitude spectrum (not the power spectrum). There is no difference in the amplitudes of either the noise or the signal between this strategy and the one suggested by the reviewer. This can be appreciated from figure 1 (below). Here, a sine wave was corrupted by noise and the spectrum computed. Then, segments of the 100-Hz signal were averaged and the spectrum calculated. The lower graph shows that both spectra have identical magnitudes. When the reviewer's method is used, the duration of the signal is reduced. As a result, frequency resolution suffers and it is no longer possible to separate closely spaced frequency components. Hence, we chose not to use the suggested method.

Fig. 1. Top: A 100-Hz noisy sine wave. **Bottom:** A direct FFT of the noisy sine wave produced the blue spectrum. The red spectrum was obtained by averaging segments of the noisy signal, and then doing an FFT. The spectra are identical except for the difference in frequency resolution.

A previous comment from one reviewer was that Fig. 2b-d aren't terribly convincing. Maybe averaging over each cycle of the envelope (at 2fm) could permit to see the effect on the temporal trace and not only on the power spectrum.

As outlined above, there is no difference between the strategy proposed by the reviewer and the one that we used, except that frequency resolution suffers when the reviewer's method is used. However, we followed the suggestion of displaying additional cycles, and found that this makes it considerably easier to appreciate the differences between the BM and the OoC potentials. In addition, low-pass filtered waveforms were included to show the envelope.

Finally, the author could low-pass filter the signal in Fig. 2b and d (let's say at 5 kHz) and also display the filtered signal below the raw traces to represent visually the envelope. This analysis could also be done for Fig.6.

Done.

b) I agree that averaging is not going to change the power spectrum which was most probably computed over the whole 100 ms segment. But was the power spectrum calculated over the averaged temporal trace or were power spectra calculated over each single trace and then averaged?

We calculated all spectra from the averaged temporal signal.

c) Was there any window used to compute the power spectrum? It seems that some peaks are wide. If the authors have used integer numbers of cycles (which seems to be the case), all the power at the primaries and at the combination tones should be contained in a single point in the power spectrum.

No window was used. Since the stimuli have several frequency components, it is not always possible to guarantee an integer number of points per cycle for every frequency component. As a result, the peaks are sometimes broader than they would otherwise have been. Note that this in no way affects the validity of the data or their interpretation.

d) Were the power spectra down-sampled for representation?

The spectra in the new figure 3c and e were down-sampled (since the peaks were otherwise so narrow that they became difficult to see).

e) It is difficult to see ticks in almost all figures (especially the log ones).

We agree. The ticks were made longer throughout.

3) Presentation of the new paradigm. As the authors argue, the use of the 3-tone stimulus is novel and enables probing the auditory nonlinearity cleverly. It might be worth making a figure with Fig. 1a and Fig. 1b only in order to present this new stimulus and discuss the expected results. The author should also present the amplitude of the 2fm envelope in Fig. 1b (see point 1).

The previous Figure 1a and b is now a separate figure, and the amplitude at 2fm is included.

I would write the equation of the 3-tone stimulus there and not in the "Mechano-electrical transduction channels generate envelope-tracking responses" section that comes too late in the paper. This is the stimulus that was used all along the study and not only in the patch-

clamp experiments. I would also include the discussion about the Taylor expansion because it is general to all approaches and gives useful intuition about the results.

We inserted the equation at the beginning of the results section along with the Taylor expansion.

4) Figure 3 and 6. It is probably important to show data on a linear frequency scale since it was represented in this way in the other figures. This is necessary to support the following statement related to Fig. 3: “High-frequency distortion products were however present on both structures.” At the present time, it is impossible to tell whether there are distortions around the primaries because the scale is too small to distinguish anything. Alternatively, a zoom of the spectra around the primaries could be shown but a linear axis would be more favorable to compare data between figures.

We chose the log display in order to highlight the low-frequency noise, which was a primary concern for the reviewer. With a linear scale, the low-frequency part of the OCT data becomes difficult to see. The new version of the figure includes insets with zoomed spectra.

5) Consistency. The authors should try to stay consistent between figures and show the same number of cycles of the envelope in each figure. Again, this is important to compare figures. Let’s say that the authors want to show a bit more than 1 cycle at fm as they did in Fig. 1a and in Fig. 2b-d which is good to see the envelope and the primaries. In contradistinction, Fig. 1d shows about 10 cycles and Fig. 6b shows 9 cycles.

We changed the figures to display a similar number of cycles (since displaying fewer cycles in Fig. 6b made the baseline less visible, we increased the number of cycles in the other figures). As mentioned above, this change made it easier to appreciate the differences between the response waveforms in Fig. 3b and d.

6) Patch-clamp and simulations. Why did the authors use different frequencies in patch-clamp and simulation experiments? I understand that it would not be possible to use 15 kHz in voltage-clamp recordings but why not in simulations using the frequencies used in experiments? There is no temporal component in the described model, so stimulation at 15 kHz or at 15 Hz will give the exact same result. It should also be made clear to the reader that the author considered the mechano-transduction channels as infinitely fast in their model.

Since the bulk of the experiments were made with high-frequency stimuli, we chose to use such stimuli in the model. As pointed out by the reviewer, this makes no difference to the results. The legend for Fig. 7 now states that the model lacks temporal parameters, hence assuming channels are infinitely fast.

Would it be also possible to show the cubic distortions in the patch-clamp experiment and in the model (the same ones as in Fig. 5)? This could be added in a supplementary figure. Also to clarify that distortions arise for a sufficient stimulus level, it would be interesting to show (maybe in an additional supplementary figure) the relative amplitude of the envelope for low (<1nm, when there is no distortion), medium (around 10 nm, when the relative amplitude of distortion is large) and large (>100 nm, when distortions saturate) displacements X for a given X0 (let’s say at X0=7 nm where distortions are the highest).

We added supplementary material showing the amplitude of the $2f_1$ - f_2 distortion product in the patch-clamp data, as well as the corresponding model output for different bundle offsets and center-tone phases.

Minor points

Line 169: “but this model also predicts responses not observed experimentally, such as frequency components centered on the second harmonic of the stimulus.” It is difficult to judge because the authors don’t show the spectrum around these frequencies.

Expanding the frequency range of the graphs makes the low-frequency parts more difficult to see, so we removed this statement.

Line 187: “2-dB loss of auditory sensitivity” This accuracy seems quite remarkable. Is this a typo? If not, please explain in the Methods how the auditory threshold was measured and what the accuracy of the method was.

The CAP was measured using an attenuator with 1-dB steps, and an averaging digital oscilloscope. The measured thresholds were 2 dB different. We did not mean to make claims about extreme accuracy, but these are the values returned from the measurement. In experienced hands, CAP thresholds are however highly reproducible.

Line 442: what is the typical cutoff frequency of the electrodes?

Typical values range between 1 and 5 kHz, and this information is now included in the methods section.

Figure 1: The frequencies used in Fig. 1a are the one used for in vivo recordings in Guinea pigs which is quite confusing because the frequencies used for Human data are different. The author should try to be consistent and present the stimulus that was actually used in this experiment.

As recommended above, we put the stimuli in a separate figure. We also removed the scale bars of the schematic to make it completely generic, and the text now uses the terminology that the reviewer suggested throughout.

It is also important to show temporal traces of same length in Fig.1a and 1d to compare directly the data (the stimulus and the response) and have a sense of the frequency that carries the distorted signal. After averaging (see previous point 2.a), it should be quite clear that the “center phase 0” trace oscillates at f_m and the “center phase 90” oscillates at $2f_m$ (we already see in the temporal trace that the red trace oscillates at twice the frequency of the blue trace but averaging should make it even more obvious).

We updated the figures and we hope the waveforms are now easier to see.

Figure 3: It should be made a bit clearer that measurements at the RW are electrical. To avoid confusion, the spectrum for RW electrical potentials could be represented in another color (and maybe another color than the blue/red pair that was used for the phase of the 3-tone stimulus).

We replotted the figure using a different color.

It would also facilitate the understanding of the figure if the authors could show RL, BM and RW for both sound intensities (in the present manuscript, RW electrical potentials are only

shown for 74 dB). If the authors don't have the data, it would help comparison between sound intensities to present RL and BM on top of each other. Finally, writing the sound intensity on the figure itself could simplify its understanding.

We did not collect RW data at 94 dB SPL, only the mechanical ones. We tried plotting the traces on top of each other but found that clarity suffers, since it becomes difficult to separate the different traces. To aid the reader in the comparison, we replotted the graphs with grid lines, and we added the sound pressure level to the graphs.

Reviewer #3 (Remarks to the Author).

The authors have addressed my concerns.

Reviewers' comments:

Reviewer # 2 (Remarks to the Author):

The authors satisfactorily addressed all my concerns. The paper is now much more complete and accessible to a wider audience. I believe that it will make an important contribution to the field.